# Object-X: Learning to Reconstruct Multi-Modal 3D Object Representations

**Gaia Di Lorenzo**[1] **Federico Tombari**[2] **Marc Pollefeys**[1,3] **Daniel Barath**[1,2]

[1]ETH Zurich    [2]Google    [3]Microsoft

gdilorenzo@ethz.ch

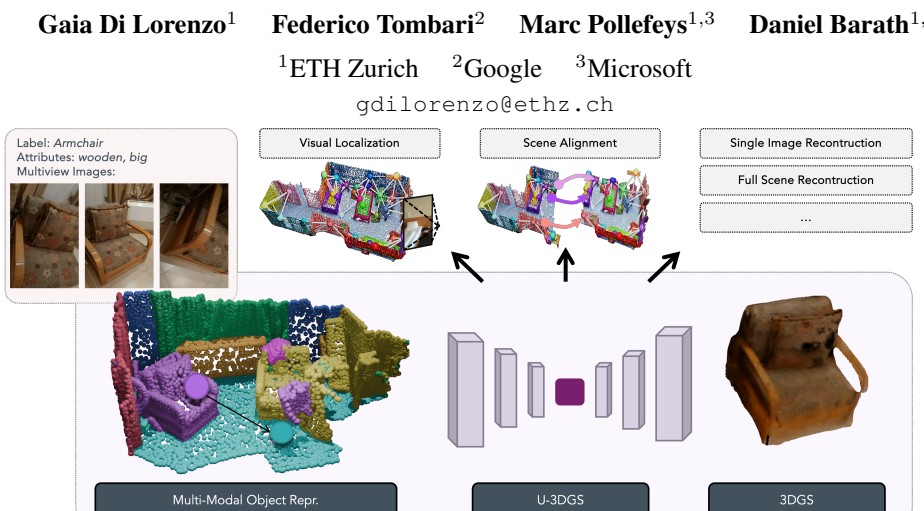

Figure 1: ***Object-X*** learns object-centric embeddings from an input object segmentation of a 3D scene reconstruction. The embeddings learned from multi-modal data (e.g., mesh, images, text descriptions) enable fast 3D Gaussian Splat reconstruction via a specifically trained decoder, and other downstream tasks operating directly in the latent space, such as localization and scene alignment. *Object-X* allows for representing the scene as a set of object descriptors without having to store storage-heavy representations like point clouds and image databases, while providing similar functionalities.

## Abstract

Learning effective multi-modal 3D representations of objects is essential for numerous applications, such as augmented reality and robotics. Existing methods often rely on task-specific embeddings that are tailored either for semantic understanding or geometric reconstruction. As a result, these embeddings typically cannot be decoded into explicit geometry and simultaneously reused across tasks. In this paper, we propose *Object-X*, a versatile multi-modal object representation framework capable of encoding rich object embeddings (e.g., images, point cloud, text) and decoding them back into detailed geometric and visual reconstructions. *Object-X* operates by geometrically grounding the captured modalities in a 3D voxel grid and learning an unstructured embedding fusing the information from the voxels with the object attributes. The learned embedding enables 3D Gaussian Splatting-based object reconstruction, while also supporting a range of downstream tasks, including scene alignment, single-image 3D object reconstruction, and localization. Evaluations on two challenging real-world datasets demonstrate that *Object-X* achieves high-fidelity novel-view synthesis comparable to standard 3D Gaussian Splatting, while significantly improving geometric accuracy. Moreover, *Object-X* achieves competitive performance with specialized methods in scene alignment and localization. Critically, our object-centric descriptors require 3-4 orders of magnitude less storage compared to traditional image- or point cloud-based approaches, establishing *Object-X* as a scalable and highly practical solution for multi-modal 3D scene representation. The code is available at https://github.com/gaiadilorenzo/object-x.

39th Conference on Neural Information Processing Systems (NeurIPS 2025).

# 1 Introduction

Robust 3D scene understanding, incorporating geometric, visual, and semantic information, forms a cornerstone for advances in robotics, augmented reality (AR), and autonomous systems (17; 12; 2). A key goal is to develop 3D representations that are not only accurate but also compact, efficient, and flexible enough to integrate multiple sensor modalities and support diverse downstream tasks.

Traditional 3D representations, often relying on explicit geometry such as dense point clouds (4; 20) or meshes (18), alongside collections of images, tend to incur prohibitive storage and computational costs. More recently, implicit neural and Gaussian representations, notably Neural Radiance Fields (NeRF) (16) and 3D Gaussian Splatting (3DGS) (11), have achieved state-of-the-art results in synthesising novel views from images, jointly encoding geometry and appearance. However, these methods typically generate a monolithic, scene-level representation, primarily driven by visual input. As a consequence, they inherently lack object-level modularity, making it difficult to reason about individual objects, efficiently incorporate other modalities (e.g., text, semantics), or easily use the representation for object-level tasks beyond rendering or 3D reconstruction.

To address the need for modularity, object-centric approaches, such as 3D scene graphs (1), have gained traction. Such methods decompose a scene into a collection of objects and their relationships, often associating a learned embedding with each object. Such embeddings have proven effective for abstract, object-level tasks, including cross-modal localization (15), scene retrieval (24), and 3D scene alignment (23). However, a critical limitation persists: existing object embeddings are generally learned for specific tasks and cannot be decoded to reconstruct the explicit, high-fidelity appearance and geometry of the object they represent. This forces systems to retain the original, high-bandwidth source data (images, point clouds, meshes) alongside the learned embeddings, undermining the goals of creating a compact, self-contained, and truly versatile object representation.

In this work, we bridge this crucial gap by introducing *Object-X*, a framework for learning rich, multi-modal, object-centric embeddings that are simultaneously suitable for downstream tasks *and* decodable into explicit, high-quality 3D representations. *Object-X* geometrically grounds multiple input modalities pertaining to an object within a 3D voxel structure, fusing this with semantic attributes (like class labels or object descriptions) to learn a compact, latent embedding. Crucially, we design a decoder that uses this embedding to predict the parameters of a set of 3D Gaussians, enabling high-fidelity, object-level rendering and geometry extraction via 3D Gaussian Splatting. The same learned embedding can be directly leveraged for diverse downstream tasks.

Our experiments on challenging, real-world datasets demonstrate that *Object-X* supports high-fidelity novel-view synthesis comparable to, and geometric reconstructions superior to, standard 3DGS, while also achieving competitive performance on scene alignment and localization tasks. Critically, *Object-X* reduces storage requirements by 3-4 orders of magnitude compared to storing the underlying point clouds or images, while offering similar functionalities. Our main contributions are:

1. A novel framework for learning compact, multi-modal, object-centric embeddings that can be decoded into high-fidelity geometry and appearance, parameterized by 3D Gaussians.
2. The demonstration that a *single*, unified embedding supports both high-quality 3D reconstruction (encompassing novel view synthesis and detailed geometry) and performs competitively on diverse downstream tasks, such as scene alignment and visual localization.
3. Significant storage reduction (3-4 orders of magnitude) compared to explicit representations as the decodability of our embeddings obviates the need to store raw images or point clouds.

# 2 Related Works

Understanding 3D scenes is a fundamental problem in computer vision, with applications spanning robotics, augmented reality, and 3D content creation. Numerous representations have been proposed to capture the complexities of 3D environments, each offering different trade-offs between accuracy, efficiency, storage, and ease of use. Our work, *Object-X*, builds on and connects several key areas by proposing a novel object-centric embedding strategy.

**3D Representations.** Traditional methods for representing environments include point clouds (4), meshes (20), and voxel grids (18; 38; 34). Point clouds offer a direct representation of geometry but lack structure and demand significant storage for detailed scenes. Meshes provide structure by connecting points with polygons, facilitating efficient rendering and geometric operations, though

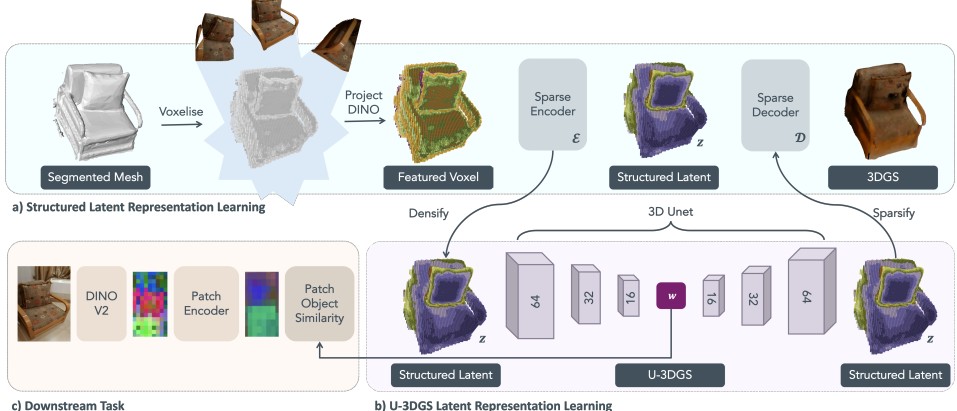

Figure 2: **Overview of *Object-X***, learning object embeddings to reconstruct 3D Gaussians and support other tasks such as visual localization (15). (a) The method takes a mesh or point cloud of an object along with posed images observing it. The canonical object space is voxelized based on object geometry, and DINOv2 features extracted from the images are assigned to each voxel. This produces a $64^3 \times 8$ structured latent (SLat) representation (31). (b) The SLat is further compressed into a $16^3 \times 8$ U-3DGS embedding using a 3D U-Net. The embedding is trained with a masked mean squared error loss to ensure accurate reconstruction of the SLat, which in turn enables decoding into 3D Gaussians using standard photometric losses. (c) Additional task-specific losses, such as those for visual localization (15), can be incorporated to optimize the embedding for multiple objectives.

the meshing process can lead to detail loss, especially when optimizing for storage. Voxel grids discretize 3D space, but suffer from high memory requirements at resolutions needed for fine detail. These representations are often accompanied by posed image datasets for applications such as visual localization, which introduces the overhead of managing and storing large image collections. The pursuit of compact and versatile representations motivates our work on learnable object embeddings.

Recently, neural implicit representations have gained attention. Neural Radiance Fields (NeRF) (16) represent scenes with MLPs, enabling photorealistic novel view synthesis but can be computationally intensive. 3D Gaussian Splatting (3DGS) (11) presents a compelling alternative, using a collection of 3D Gaussians for high-quality, real-time rendering. While methods like MV-Splat (3), Depth-Splat (32), NoPoSplat (35), and alternatives learn a monolithic 3DGS model for an entire scene from images, we focus on an object-centric paradigm. We learn compact, multi-modal embeddings for individual objects, which can then be decoded into object-level 3DGS parameters. Although recent works have explored 3DGS modifications for editing (37) or compression (25), they typically operate on or refine an existing 3DGS scene. In contrast, *Object-X* learns fundamental object embeddings that serve not only as a source for 3DGS reconstruction but also as versatile descriptors for various other downstream tasks, offering a more holistic object representation.

**3D Scene Graphs.** Scene graphs (10) provide a structured representation by capturing objects, their attributes, and interrelations. 3D scene graphs (1) extend this concept to 3D, integrating semantics with spatial and camera information. They have proven useful for tasks like scene alignment (23), retrieval (15; 24), and task planning (5). The construction of 3D scene graphs has been streamlined by recent advances in object detection and relationship prediction, with tools such as MAP-ADAPT (38), OpenMask3D (26), and ConceptGraphs (5). Existing scene graph methodologies often associate nodes with learned embeddings tailored for specific tasks (e.g., alignment, retrieval). However, a key limitation is that these embeddings typically lack a generative or reconstructive capability; they cannot be decoded back into explicit object geometry or appearance. This necessitates retaining the original sensor data (images, point clouds) alongside the graph, undermining compactness. *Object-X* addresses this gap by learning rich, multi-modal object embeddings that are explicitly designed to be decodable into high-fidelity 3DGS representations. Furthermore, these same embeddings retain strong descriptive power, enabling competitive performance on downstream tasks like localization and alignment without requiring task-specific modifications. This dual capability – reconstructive and descriptive – is a core contribution of our work, offering a pathway to leverage the semantic richness while also providing access to explicit 3D object representations.

**3D Generative Methods** for content creation have rapidly advanced, with 3DGS (11) emerging as an expressive and efficient primitive. Several methods leverage 2D distillation from text or images to generate 3D 3DGS scenes (27; 21). More recent works explore structured latent spaces for improved scalability and control in generation. For instance, Trellis (31) uses a unified Structured LATent (SLat) representation, decodable into various 3D forms including Gaussians, for large-scale object generation. L3DG (22) employs a latent diffusion framework with VAEs for efficient sampling and 3DGS rendering, while DiffGS (39) introduces a diffusion model for controlling Gaussian parameters.

These methods focus on *de novo* generation or generation from abstract inputs (e.g., text prompts, style images), showcasing the potential of learned latent spaces for 3D content creation. *Object-X* shares the goal of leveraging learned latents but differs in its main objective. Instead of open-ended generation, our focus is on learning compact and versatile embeddings from *observed multi-modal data*. The key is to create embeddings that capture an object geometry and appearance for reconstruction, while also being discriminative enough for tasks like localization and alignment. Thus, we emphasize robust representation learning from captured data for multi-task utility, rather than pure synthesis.

## 3 Learning Versatile Object Embeddings

We propose *Object-X*, taking a reconstructed scene with a 3D object segmentation as input and learning a compact and descriptive embedding for each object from their associated multi-modal data (e.g., images, point cloud). To achieve this, we process each 3D object through the following steps:

1) **Extract Structured Latent Representation (SLat):** We first process the input data for an object. We voxelize the object into a canonical 3D grid and aggregate local image features within these voxels via multi-view projection, inspired by (31). A 3D encoder then transforms these voxel-aligned features into a *structured set* of latent vectors (the SLat), which represents the object's initial rich encoding.

2) **Project SLat to an Unstructured Embedding:** We then learn to compress SLat into a *dense and unstructured* latent representation of a fixed, significantly smaller dimension. Besides compression loss, other objectives are also considered during training to facilitate downstream tasks, e.g., object retrieval. This embedding, which we term the U-3DGS Embedding[1], becomes our core storage unit and descriptor for an object.

3) **Decode the U-3DGS Embedding to 3DGS:** One key application of the U-3DGS embedding is its decodability. For reconstruction, we map this dense embedding back to an SLat representation and then decode this into a full 3D Gaussian Splat model for the object.

Below, we detail the formation of the SLat representation. We then describe its compression into the versatile U-3DGS embedding. Following this, we explain the decoding mechanism that enables 3DGS-based reconstruction from this embedding. The utility of U-3DGS for other direct downstream applications will be demonstrated in subsequent sections. Fig. 2 summarizes the overall pipeline.

### 3.1 Structured Latents from Multi-View Images

Let a set of object instances $\mathcal{O}$ be given, where each object $o = (\mathcal{P}, \mathcal{I}, \mathcal{M}, \mathcal{A}, \dots) \in \mathcal{O}$ is associated with a multi-modal input, such as multi-view images ($\mathcal{I}$), instance masks ($\mathcal{M}$), point clouds ($\mathcal{P}$) and other attributes ($\mathcal{A}$). For each object $o$, we extract an object-specific latent embedding $\mathbf{w}_o$. The scene representation is defined as $\mathcal{W} = \{\mathbf{w}_o\}_{o \in \mathcal{O}}$, enabling lightweight storage compared to dense point clouds or images. The goal is to learn mapping $\mathcal{O} \mapsto \mathcal{W}$, where $\mathcal{W}$ allows decoding into 3D objects. Here, we focus on modalities $\mathcal{P}$ and $\mathcal{I}$ to learn the reconstructable embedding. Other modalities will be discussed when training also for downstream applications in the next section. We denote the resulting latent object embedding as $\mathbf{w}$ which is the central representation used throughout the paper.

**Voxelization and Feature Extraction.** Given multi-view images of an object $o$, we first voxelize its canonical 3D space into an $N \times N \times N$ grid (e.g., $64^3$). For each voxel $p_i$ that intersects the surface of the object, we project $p_i$ into each image to retrieve localized features. Similar to (31), we employ a pre-trained DINOv2 feature encoder (19) on masked object images. Averaging these image-level features yields a per-voxel feature $f_i \in \mathbb{R}^D$. Concatenating over all *active* voxels forms a *sparse, voxel-aligned feature set* as $f = \{(f_i, p_i)\}_{i=1}^{L}, L \ll N^3$.

---

[1]Named for its designed decodability into 3D Gaussian Splats, though it serves broader purposes.

**Structured Latent Representation (SLat).** We convert $f$ into a structured latent representation $z = \{(z_i, p_i)\}_{i=1}^{L}$ via a 3D encoder $\mathcal{E}$ as follows:

$$z = \{(z_i, p_i)\}_{i=1}^{L} = \mathcal{E}(f), \ z_i \in \mathbb{R}^C, \ p_i \in \{0, \ldots, N-1\}^3.$$

Each voxel $p_i$ is paired with a local latent $z_i$ capturing shape and appearance. By preserving the sparse grid structure, $z$ offers geometric grounding (through $p_i$) and localized feature encoding (through $z_i$).

## 3.2 Compression to a Dense Embedding

Although $z$ is sparse relative to raw 3D data, it can still be large at high resolutions. Capturing fine details requires dense grids, which remain memory intensive. Moreover, downstream tasks favor a compact embedding for each object, rather than a collection of per-voxel vectors. Thus, we learn a mapping to compress $z$ into a fixed-size vector $\mathbf{w} \in \mathbb{R}^d$, where $d$ is small and independent of the size of the object at hand as $\mathbf{w} = f_{\text{U-3DGS}}(z)$, where $\mathbf{w}$ is an Unstructured 3D Gaussian Splatting (U-3DGS) embedding. We implement $f_{\text{U-3DGS}}$ using a 3D network (similar to a U-Net) that first organizes $z$ into a dense $N \times N \times N \times C$ tensor (with zero-filling for inactive voxels), then, downsamples and encodes it to $\mathbf{w}$.

**Masked MSE for Compression Learning.** We supervise $f_{\text{U-3DGS}}$ by requiring that the dense embedding $\mathbf{w}$ *can be decoded back* into the original structured representation $z$ (or a close approximation). Specifically, we introduce a decompression function $f_{\text{decomp}}$ that maps $\mathbf{w}$ back to a predicted $\hat{z} = \{\hat{z}_i, p_i\}_{i=1}^{L}$. We then encourage $\hat{z}_i \approx z_i$ under a masked mean-square-error (MSE) or L1 loss that focuses on occupied voxels as $\hat{z} = f_{\text{decomp}}(\mathbf{w})$, and the objective includes

$$\mathcal{L}_{\text{compress}} = \frac{1}{N^3} \sum_{i=1}^{N^3} \Big[ M_i \|\hat{z}_i - z_i\|^2 + \tfrac{1-M_i}{w} \|\hat{z}_i - z_i\|^2 \Big],$$

where binary mask $M_i$ indicates if voxel $p_i$ is occupied, and $w$ is a down-weighting factor for non-occupied regions.

## 3.3 Decoding to 3D Gaussian Splats

In this section, we use the learned object embeddings $\mathbf{w}$ to reconstruct the object into a 3DGS representation. As discussed in the previous section, we can obtain the reconstructed structured latent representation $\hat{z}$ from $\mathbf{w}$ by applying decompression network $\hat{z} = f_{\text{decomp}}(\mathbf{w})$.

**Deterministic Autoencoder for 3DGS.** We next train a decoder $D_{\text{GS}}$ that takes in the reconstructed structured latents $\hat{z}$ (obtained from $\mathbf{w}$) and outputs a set of 3D Gaussian Splat parameters as follows:

$$\Theta = \mathcal{D}_{\text{GS}}(\hat{z}), \quad \Theta = \Big\{ (x_i, s_i, q_i, \alpha_i, c_i) \Big\}_{i=1}^{M}. \tag{1}$$

Each Gaussian is specified by position $x_i$, scale $s_i$, rotation $q_i$, opacity $\alpha_i$, and color $c_i$. We train $D_{\text{GS}}$ by rendering these Gaussians from multiple viewpoints and minimizing image-space reconstruction losses (e.g., L1, SSIM, and LPIPS) against ground-truth images of the object using loss $\mathcal{L}_{\text{render}} = \lambda \mathcal{L}_{\text{L1}} + (1 - \lambda)\big[1 - \mathcal{L}_{\text{SSIM}}\big] + \mathcal{L}_{\text{LPIPS}}$. Since $\hat{z}$ can accurately encode fine object details, $D_{\text{GS}}$ learns to produce high-fidelity splats.

**Voxel-Level Offsets.** For spatial alignment, the center of each Gaussian is computed as $x_i = p_i + \tanh(o_i)$, where $o_i$ is an offset predicted by $D_{\text{GS}}$ and $p_i$ is the voxel location. This ensures positions remain near the coarse voxel layout, but can adjust locally for more precise fits.

**Training.** The proposed pipeline is end-to-end trainable, learning a compact embedding space in a single training pass. The optimization jointly minimizes the reconstruction loss, ensuring accurate recovery of the SLat representation from the U-3DGS embedding, and photometric loss, learning the mapping from SLat features into 3DGS that can be used for NVS and surface reconstruction.

## 3.4 Learning Auxiliary Tasks

The U-3DGS embedding, primarily learned to capture object appearance and geometry for high-fidelity reconstruction, also serves as a potent foundation for various downstream auxiliary tasks such as visual localization (15) and 3D scene alignment (23; 24) (see Fig. 3). To further enhance performance on such tasks, we augment the pre-trained U-3DGS representation by integrating information from other relevant modalities. This augmentation involves incorporating features

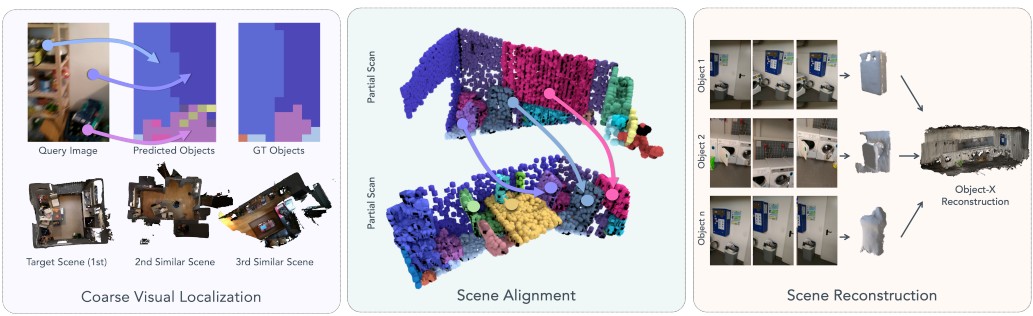

Figure 3: The proposed *Object-X* learns per-object embeddings that are beneficial for a number of downstream tasks, besides object-wise 3DGS reconstruction, such as cross-modal visual localization (15) (via image-to-object matching), 3D scene alignment (23) (via object-to-object matching), and full-scene reconstruction by integrating per-object Gaussians primitives.

from auxiliary data sources like textual descriptions, object relationships, or broader scene context (which can be derived, for example, from a 3D scene graph structure (23)). Each auxiliary modality is processed by its own dedicated encoder (e.g., a CLIP-based model with a projection head for text features), following (23). The resulting feature vectors from these auxiliary encoders are then concatenated with the original U-3DGS embedding to form a richer, multi-faceted representation for the object. The training strategy for these augmented representations, leveraging the pre-trained U-3DGS encoder and decoder, proceeds in two main stages after the initial U-3DGS pre-training:

**Auxiliary Encoder Training with Frozen Core:** Initially, the pre-trained U-3DGS encoder and decoder (responsible for appearance and geometry) are kept frozen. The newly introduced encoders for the auxiliary modalities are trained. In this stage, learning is guided only by the task-specific loss $\mathcal{L}_{\text{task}}(\mathbf{w}_{\text{concat}})$, where $\mathbf{w}_{\text{concat}}$ is the full concatenated embedding (U-3DGS + auxiliary features). For example, $\mathcal{L}_{\text{task}}$ may be a contrastive loss for localization. This ensures that the new auxiliary features are learned in a way that remains compatible with, and does not corrupt, the reconstructive capabilities of the core U-3DGS representation.

**Joint Fine-tuning:** After the auxiliary encoders have been trained, all network components are unfrozen. The entire ensemble is then fine-tuned end-to-end using a combined objective:

$$\mathcal{L}_{\text{aux}} = \mathcal{L}_{\text{task}}(\mathbf{w}_{\text{concat}}) + \lambda_{\text{recon}}\mathcal{L}_{\text{recon}}(\mathbf{w}_{\text{U-3DGS}}), \tag{2}$$

where $\mathcal{L}_{\text{recon}}$ is the reconstruction loss also used for pre-training the U-3DGS embeddings, applied using the U-3DGS decoder on the corresponding $\mathbf{w}_{\text{U-3DGS}}$. $\lambda_{\text{recon}}$ balances these two objectives. This stage allows for mutual adaptation of all parts of the representation, further optimizing for both the specific auxiliary task and the foundational 3D reconstruction quality.

Through this process, *Object-X* learns to effectively fuse intrinsic object properties (geometry, appearance via U-3DGS) with extrinsic or contextual information (text, relationships via auxiliary encoders). We will demonstrate this approach by training our model for visual localization (15), and subsequently evaluate its zero-shot or fine-tuned performance on related tasks like 3D scene alignment, showcasing the versatility and robustness of the learned augmented embeddings.

## 4 Experiments

Next, we will provide experiments on various tasks benefiting from *Object-X*. Ablation studies, more visuals, and detailed descriptions of baselines are provided in the supplementary material.

**Mesh extraction.** To evaluate the geometric accuracy, we extract a triangle mesh from the optimized 3D Gaussians, following the procedure from 2DGS (41). We first render depth maps from different viewpoints. These depths are fused using the Truncated Signed Distance Function (TSDF) integration in Open3D (40). Finally, a triangle mesh is extracted using Marching Cubes (14).

**Implementation details.** All experiments are conducted on a machine with an A100 GPU with 80GB of RAM. During sparsification, a threshold of 0.5 is applied to the predicted occupancy. The mesh is constructed using a voxel size of 0.015 and an SDF truncation value of 0.04.

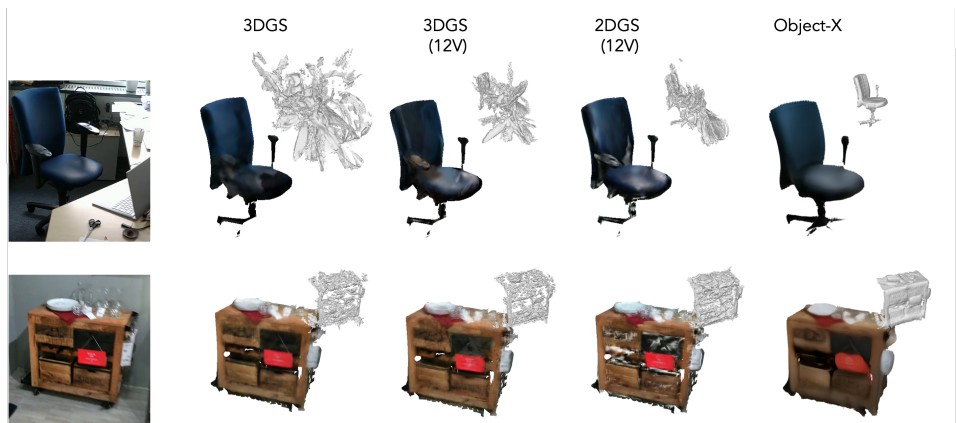

Figure 4: **Object reconstructions.** Each row shows an input object (left) and its reconstruction obtained by, from left to right: (i) 3DGS (28) optimized on all images, (ii) 3DGS or (iii) 2DGS (41) using only 12 multi-view images, and (iv) *Object-X*. For each method, we present a rendered image from the reconstructed 3D Gaussians and the corresponding mesh.

Table 1: **3DGS Object reconstruction** photometric quality, geometric accuracy, runtime, and storage efficiency on 3RScan (9) and ScanNet (4). We compare *Object-X* with baselines that store objects as a set of 12 (3RScan) or 4 images (ScanNet) and reconstruct 3D Gaussians at test time using 3DGS (28), 2DGS (41), and DepthSplat (32). As a reference, we report the results of 3DGS, optimizing directly on all dataset images. We report NVS scores (SSIM, PSNR, LPIPS), geometric accuracy (Accuracy, Completion, and F1 score at a 0.05 m threshold), per-object run-time (secs), and storage (MB). We do not show geometric accuracy for DepthSplat as it failed mesh reconstruction.

| | Method | SSIM ↑ | PSNR ↑ | LPIPS ↓ | Acc.@0.05 ↑ | Compl.@0.05 ↑ | F1@0.05 ↑ | Time (s) ↓ | Storage (MB) |
|---|---|---|---|---|---|---|---|---|---|
| **3RScan** | 3DGS (28) | 0.956 | 34.009 | 0.051 | 33.75 | 73.38 | 41.41 | 58.461 | 32.74 |
| | 3DGS (12V) (28) | 0.944 | **31.613** | 0.072 | 33.72 | **77.81** | 44.35 | 58.461 | 6.71 |
| | 2DGS (12V) (7) | 0.932 | 29.613 | 0.093 | 26.55 | 67.69 | 35.67 | 84.280 | 6.71 |
| | DepthSplat (12V) (32) | 0.619 | 21.669 | 0.304 | - | - | - | 0.491 | 6.71 |
| | *Object-X* | **0.953** | 30.981 | **0.065** | **80.22** | 77.80 | **77.80** | **0.051** | **0.14** |
| **ScanNet** | 3DGS (28) | 0.975 | 38.138 | 0.032 | 33.81 | 79.13 | 43.00 | 129.02 | 270.97 |
| | 3DGS (4V) (28) | 0.949 | 30.754 | 0.059 | 36.95 | 82.37 | 48.18 | 129.02 | 55.26 |
| | 2DGS (4V) (7) | 0.945 | 29.604 | 0.073 | 26.96 | 74.85 | 37.10 | 169.65 | 55.26 |
| | DepthSplat (4V) (32) | 0.832 | 26.152 | 0.134 | - | - | - | 0.138 | 55.26 |
| | *Object-X* | **0.966** | 31.563 | **0.047** | **88.66** | **90.08** | **89.09** | **0.033** | **0.14** |

**Datasets.** The *3RScan* dataset (9) consists of 1,335 annotated indoor scenes covering 432 distinct spaces, with 1,178 scenes (385 rooms) used for training and 157 scenes (47 rooms) reserved for validation and testing. The dataset provides semantically annotated 3D point clouds, with certain scenes captured over extended periods to reflect environmental changes. Scene graph annotations are available from (28). Since the test set lacks such annotations, we reorganized the original validation split, allocating 34 scenes (17 rooms) for validation and 123 scenes (30 rooms) for testing. Objects without available images were removed to ensure a consistent evaluation.

*ScanNet.* To evaluate generalization, we test on ScanNet (4) *without* training our model on it. Since ScanNet does not provide scene graph annotations, we apply SceneGraphFusion (30) on RGB-D sequences to generate 3D instance segmentations and object relationships (used for auxiliary tasks). This allows us to assess robustness to errors in 3D instance segmentation in the practical setting. Compared to 3RScan, ScanNet captures RGB-D sequences at a higher frame rate with minimal motion between consecutive frames. To ensure diverse viewpoints, we sample one image every 25 frames. We use 77 test scenes from the split defined in (15). Scenes, where SceneGraphFusion fails to generate annotations, are excluded. As in 3RScan, objects without associated images are discarded. The test split, along with its annotations, will be publicly released.

**1. Object Reconstruction.** First, we evaluate the *Object-X* decoder in terms of storage efficiency, geometric fidelity, and visual quality on the object reconstruction task.

*Baselines.* 3DGS (11) serves as a high-fidelity baseline, representing each object as a set of 3D Gaussians. While this approach captures fine details, it requires substantial storage, as every object is

represented by a set of Gaussians. We provide results for *3DGS (12 views)* that stores each object as 12 images captured from different viewpoints. During reconstruction, 3DGS is applied to recover the 3D Gaussians from these views. This reduces storage compared to full 3DGS but introduces a trade-off: reconstruction takes longer, and the quality may be slightly degraded. *2DGS (12 views)* (7) follows the same 12-image storage strategy but employs 2DGS (41) instead of 3DGS. Both 2DGS and 3DGS leverage the segmented mesh as initialization. Default parameters are used. *DepthSplat (12 views)* (32) also relies on 12 stored views but reconstructs objects using DepthSplat, a fast feed-forward network. We use the pre-trained model provided by the authors, which was not trained on 3RScan. Since we train on 3RScan, comparisons on this dataset may be unfavorable to DepthSplat. However, we also evaluate on ScanNet, where neither DepthSplat nor our method has been trained.

For the baselines, we select $k$ frames that maximize viewpoint diversity by applying $k$-means clustering to the positions of the cameras observing the object. In 3RScan, we use the maximum number of frames supported by DepthSplat (i.e., 12), while in ScanNet, where fewer frames are available, we limit the selection to four views per object. To ensure fairness, 3DGS/2DGS are also evaluated under this sparse-view setting ($\leq 12$ views), matching the input regime of DepthSplat and reflecting the natural sparsity of 3RScan/ScanNet. DepthSplat itself is evaluated in its intended setup on unmasked scene-level images, with object masks applied only after reconstruction, ensuring a consistent and fair comparison.

We present examples in Fig. 6, showing renderings and the reconstructed meshes. *Object-X* produces significantly smoother renderings and higher-quality meshes, whereas meshes reconstructed by baselines exhibit strong artifacts and fail to achieve accurate geometry.

*Metrics.* We evaluate our method using the standard novel view synthesis scores: PSNR, SSIM, and LPIPS. Additionally, we report standard geometric metrics: accuracy, completeness, and F1 score.

***Results on 3RScan.*** The top part of Table 1 reports the results on 3RScan. *Object-X* achieves comparable novel view synthesis quality to the baselines, with an SSIM score closest to 3DGS, the second-best PSNR score, and the best LPIPS score. *Object-X* substantially outperforms *all* baselines in geometric accuracy. Our method improves geometric accuracy by a large margin of *46 percentage points* compared to all baselines while also exhibiting good completeness. This demonstrates that the proposed U-3DGS embeddings effectively capture object geometry, accurately recovered by the *Object-X* decoder. We omit geometric results for DepthSplat (32), which failed to produce reasonable geometry. We attribute the higher geometric accuracy to the voxel-grounded latent representation, which spatially constrains and aligns the decoded Gaussians to surfaces, leading to improved consistency compared to unstructured Gaussian sets.

Our runtime is *three orders of magnitude* faster than methods relying on optimization. Moreover, our approach requires *an order-of-magnitude* less storage, as we only store a single embedding per object instead of numerous images or 3D Gaussians.

***Results on ScanNet.*** The bottom part of Table 1 reports the results on ScanNet. Note that we did not train our model on this dataset and used the model trained on 3RScan. Even without training, we achieve the highest novel view synthesis scores compared to the baselines, being the closest to the reference 3DGS reconstruction. Our geometric accuracy significantly outperforms all baselines. Because ScanNet provides higher-resolution images than 3RScan, optimization-based approaches are considerably slower, even when using only 4 views. We are *four* orders of magnitude faster than 3DGS (4V) and 2DGS (4V) requiring 3 ms to reconstruct an object on average.

**2. Scene Reconstruction.** Although we do not explicitly perform scene reconstruction, we evaluate scene composition by integrating object-level reconstructions. *Object-X* composes the full scene by decoding embeddings for each object and rendering their splats jointly. While this method is highly efficient, artifacts can emerge due to missing object segmentations. We also show results for *Object-X + Opt*, where the initial scene is constructed via *Object-X*, and optimized using 3DGS on the same image set used in 3DGS (12V). This setup isolates the benefit of *Object-X* as a strong initialization while allowing to overcome the problems caused by missing objects.

Table 3 reports results on the 3RScan dataset. While *Object-X* achieves lower SSIM and PSNR compared to 3DGS (12V), it significantly outperforms *all* methods in geometric accuracy. Also, it runs two orders of magnitude faster than the baselines. With refinement (*Object-X + Opt*), our method not only matches or exceeds the photometric quality of 3DGS (12V), but also achieves the

| | Method | LPIPS ↓ | PSNR ↑ | SSIM ↑ | F1@0.05 ↑ |
|---|---|---|---|---|---|
| RGB-D | 3DGS | 0.131 | 26.58 | 0.891 | 27.01 |
| RGB-D | *Object-X* | **0.099** | **28.03** | **0.926** | **44.79** |
| RGB | 3DGS | 0.129 | 26.72 | 0.896 | 8.70 |
| RGB | *Object-X* | **0.110** | **27.27** | **0.918** | **10.53** |

**(a) Single-Image Object Reconstruction** on RGB (depth predicted by (6)) and RGB-D frames from 3RScan (9). Photometric and geometric accuracy of 3DGS vs. *Object-X*.

| Method | Modalities | | | | 10 scenes | | |
|---|---|---|---|---|---|---|---|
| | $\mathcal{P}$ | $\mathcal{I}$ | O | 3DGS | R@1 | R@3 | R@5 |
| SGLoc (15) | ✓ | ✗ | ✓ | ✗ | 53.6 | 81.9 | **92.8** |
| CrossOver (24) | ✗ | ✓ | ✗ | ✗ | 46.0 | 77.9 | 90.5 |
| *Object-X* | ✗ | ✗ | ✓ | ✓ | **56.6** | **82.2** | 91.8 |

**(b) Coarse Visual Localization** on 3RScan (9). Retrieval recall at various thresholds using various methods and input modalities.

Figure 5: Comparison of (a) object reconstruction and (b) coarse localization performance using 3DGS and *Object-X* across tasks and input modalities.

highest geometric accuracy by a large margin. Runtime remains comparable to the 12-view baselines, and significantly faster than full-scene 3DGS optimization.

While we note that full-scene reconstruction is not the primary focus of this work, we include these experiments to demonstrate the viability of composing object-level embeddings into larger-scale reconstructions, where Object-X provides competitive quality, the highest geometric accuracy, and serves as a fast initializer for subsequent refinement.

**3. Single-Image Reconstruction.** We further evaluate the flexibility of *Object-X* by reconstructing objects from a single RGB or RGB-D frame. Let us note that the *Object-X* was *not* explicitly trained for this task nor to infer unseen parts of an object. In the RGB-only scenario, we estimate monodepth using (6). Using the object mask and depth map, we lift the object pixels in 3D, obtaining a point cloud. We then voxelize it following the process described in Sec. 3.1. *Object-X* is then applied to obtain the object embedding from this input which is then fed directly into our decoder. We compare our approach to 3DGS (28) which optimizes 3D Gaussian splats based on a single masked image. Results in Table 1a show that *Object-X* outperforms 3DGS in all metrics on RGB-D and RGB inputs while also achieving significantly better F1 scores, indicating more precise and reliable reconstructions. We note that while there is still room for improvement in this task, achieving improved results demonstrate the versatility of the learned object embeddings. A promising direction, for example, is the principled integration of monocular 3D priors (33; 36; 29) could further enhance geometric fidelity without sacrificing efficiency.

Additionally, we compare *Object-X* with the recent MIDI-3D (8), a diffusion-based generative model for single-image 3D reconstruction. Unlike Object-X, which encodes metrically grounded geometry from observed data, MIDI is a *purely generative* approach that learns to synthesize plausible 3D shapes directly from RGB inputs. The results are shown in Table 2. While MIDI achieves impressive visual quality on in-distribution images similar to its training data, its performance drops significantly on out-of-distribution scenes, such as those in 3RScan, where object shapes, scales, and textures differ from common internet imagery. In these cases, MIDI often produces geometrically inconsistent outputs that remain visually plausible but lack realism.

In contrast, *Object-X* focuses on *reconstruction rather than generation*, leveraging voxel-grounded latent representations to maintain geometric consistency even under large appearance or domain shifts. As a result, it generalizes better to real-world scenes without requiring distribution-specific fine-tuning, producing reconstructions that are both structurally coherent and metrically faithful.

| Method | Accuracy@0.05 ↑ | Completion@0.05 ↑ | F1@0.05 ↑ |
|---|---|---|---|
| MIDI (8) | 29.522 | 44.342 | 34.516 |
| Object-X (RGB) | 43.480 | 57.214 | 48.397 |
| Object-X (RGB-D) | 65.780 | 61.759 | 63.223 |

Table 2: Comparison with MIDI (8) on a subset of 3RScan. Metrics are computed at a 5 cm threshold. Methods marked with * use automatic alignment without manual registration.

**4. Coarse Visual Localization.** We evaluate visual localization on 3RScan. Following the protocol from SceneGraphLoc (15), we evaluate on 123 scenes from 30 rooms in the test set. We select query images for each scene and match them against 10 candidate scenes (including the target) to determine if the correct scene can be identified. This process is repeated for every image in each room, resulting

Table 3: **Full-scene composition** on 3RScan (9). We compare *Object-X* to 3DGS (28) optimized on all unmasked images, and two 12-view baselines: 3DGS (12V) and 2DGS (12V), which optimize scenes using a subset of training images constructed by taking the union of the 12 best views selected per object. *Object-X* achieves the second highest geometric accuracy the fastest. When combined with refinement (*Object-X + Opt*), it also achieves the best perceptual quality among all methods.

| | Method | SSIM ↑ | PSNR ↑ | LPIPS ↓ | Acc.@0.05 ↑ | Compl.@0.05 ↑ | F1@0.05 ↑ | Time (s) ↓ |
|---|---|---|---|---|---|---|---|---|
| 3RScan | 3DGS (28) | 0.855 | 24.404 | 0.417 | 16.48 | 31.99 | 21.40 | 260 |
| | 3DGS (12V) (28) | **0.767** | 18.806 | 0.517 | 16.81 | 32.63 | 21.88 | 150 |
| | 2DGS (12V) (7) | 0.752 | **18.900** | 0.523 | 18.09 | 30.92 | 22.61 | 250 |
| | *Object-X* | 0.677 | 15.488 | 0.526 | 46.22 | 54.98 | 49.99 | **1** |
| | *Object-X + Opt* | 0.727 | 17.098 | **0.507** | **63.00** | **71.54** | **66.49** | 150 |

Table 4: **3D Scene Alignment** on 3RScan (9) by *Object-X*, SGAligner (23) and EVA (13). We report Mean Reciprocal Rank and Hits@K that denotes the proportion of correct matches appearing within the top $K$, based on cosine similarity. Evaluations are conducted using modalities: point cloud ($\mathcal{P}$), others ($\mathcal{O}$), and 3DGS. In constrast to the baselines, *Object-X* is used *without* training on this task.

| Method | Modalities | | | Mean RR ↑ | Hits@1 ↑ | Hits@2 ↑ | Hits@3 ↑ | Hits@4 ↑ | Hits@5 ↑ |
|---|---|---|---|---|---|---|---|---|---|
| | $\mathcal{P}$ | $\mathcal{O}$ | 3DGS | | | | | | |
| EVA (13) | ✓ | ✗ | ✗ | 0.867 | 0.790 | 0.884 | 0.938 | 0.963 | 0.977 |
| SGAligner (23) | ✓ | ✗ | ✗ | 0.884 | 0.835 | 0.886 | 0.921 | 0.938 | 0.951 |
| SGAligner (23) | ✓ | ✓ | ✗ | **0.950** | **0.923** | **0.957** | **0.974** | **0.982** | **0.987** |
| *Object-X* | ✗ | ✓ | ✓ | 0.910 | 0.864 | 0.917 | 0.948 | 0.965 | 0.975 |

in a total of 30,462 query images used for evaluation. Our method, along with SceneGraphLoc and the recent CrossOver (24), uses a ViT to extract per-patch object embeddings from the query image. For each candidate scene, a similarity score is calculated by identifying the most similar object embedding in the scene for each patch in the image. Robust voting is then performed across all patch-object similarities to determine the scene where the query image was captured. We report the scene retrieval recall at 1, 3, and 5, measuring the percentage of queries for which the correct scene is ranked within the top 1, 3, and 5 retrieved scenes. The results are shown in Table 1b. We indicate whether a method uses point cloud ($\mathcal{P}$), image ($\mathcal{I}$), other modalities like object attribute and relationship ($\mathcal{O}$), or the proposed U-3DGS embedding. Our method, leveraging 3DGS and other modalities, achieves the highest R@1 and R@3 scores.

**5. 3D Scene Alignment.** To further assess our generalization capabilities, we evaluate on a new task, Scene Alignment, on 3RScan, following the protocol from SGAligner (23). This task involves matching objects across partially overlapping scans of the same scene by comparing their embeddings. Unlike SGAligner, explicitly trained for this task using point cloud and object-level modalities, our method relies solely on the proposed *Object-X* embedding trained with reconstruction and localization losses, *without* finetuning for scene alignment. As shown in Table 4, *Object-X* performs comparably in all metrics (achieving the second highest accuracy) to the baselines tailored for this task.

## 5 Conclusion

We introduced *Object-X*, a novel framework for learning compact, versatile, multi-modal object-centric embeddings. These unique embeddings are decodable into high-fidelity 3D Gaussian Splats for object reconstruction while also serving as potent descriptors for diverse downstream tasks, such as visual localization, single-image reconstruction, and 3D scene alignment, often without task-specific fine-tuning. *Object-X* achieves excellent geometric accuracy and novel-view synthesis, comparable or superior to specialized methods, while drastically reducing storage requirements by 3-4 orders of magnitude by obviating the need for raw sensor data. Our approach effectively bridges the gap between abstract learned object representations and detailed explicit 3D models, offering a scalable and practical solution for advanced 3D understanding.

*Limitations.* Despite these advances, *Object-X* has limitations. The high degree of compression can lead to a loss of the finest details in some reconstructions, particularly for large or complex objects. Furthermore, while promising in zero-shot scenarios for tasks like single-image object reconstruction, performance does not yet consistently match that of optimized task-specific methods.

*Acknowledgements.* This work was supported by an ETH Zurich Career Seed Award.

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

# Supplementary Material

This supplementary material provides additional training details, experimental setups, visualizations, and ablation studies in support of the main paper. It is organized as follows:

1. Additional details on baseline comparisons and visualizations for object-level, image-based, and scene-level reconstruction **(Section A)**
2. Training procedures for pretraining, compression, and downstream adaptation **(Section B)**
3. Setup and evaluation details for visual localization using U-3DGS embeddings **(Section C)**
4. Scene alignment task setup, including sub-scene construction and evaluation metrics **(Section D)**
5. Single image reconstruction experiment **(Section E)**
6. Ablation results on compression, occlusion robustness, and architectural variants **(Section F)**

## A    Baselines and Visualizations

This section details the implementation and evaluation protocols for all baseline methods discussed in the main paper. We cover object-level, scene-level, and single-image reconstruction settings. To ensure a fair and rigorous comparison across all experiments, we maintain consistent supervision levels, initialization strategies, and evaluation metrics when assessing storage requirements, visual quality, and geometric fidelity.

### A.1    Object Reconstruction Baselines

All optimization-based methods, specifically 3D Gaussian Splatting (3DGS) (11) and 2DGS (41), operate on masked RGB-D input sequences. Our comparative analysis includes three primary optimization-based baselines:

1. **3DGS (Full Scene):** Utilizes *all* available images from a given scene to serve as an upper-bound reference reconstruction.
2. **3DGS ($k$V):** Employs $k$ pre-selected views that observe the target object.
3. **2DGS ($k$V):** Also uses $k$ pre-selected views observing the target object.

To ensure a fair comparison by providing identical starting conditions for all methods, we initialize Gaussian splats directly from ground-truth object meshes. The $k$ views for object-specific baselines are selected using a $k$-means clustering strategy (detailed below) to promote viewpoint diversity and ensure high object visibility. Crucially, evaluation is consistently performed on a disjoint test set of images that were *not* utilized during the training or optimization phases of any method. For both 3DGS and 2DGS, we conduct optimization for 7,000 iterations using their default hyperparameter settings. For experiments on the 3RScan (9) dataset, we use 12 views and, on ScanNet (4), which typically offers fewer views per object instance, we restrict the number of selected views, $k$, to a maximum of 4 per object.

Regarding DepthSplat (32), we evaluate using the publicly available pre-trained model. As this model was not trained on object reconstruction tasks, we apply it to the *unmasked* image and, we post-process its output by removing any splats that fall entirely outside the masked object region.

Visual comparisons are provided in Figure 6. Alongside rendered novel views, we present mesh reconstructions derived from the 3D Gaussians using the TSDF fusion technique, as proposed in (41). These visualizations demonstrate that our proposed method, *Object-X*, achieves significantly smoother novel view syntheses and more geometrically accurate mesh reconstructions compared to the baselines.

**Frame Selection Protocol.** To ensure consistent and representative view selection across all relevant experiments (object-level and scene-level $k$-view baselines), we employ a clustering-based strategy for choosing training/optimization views. From the available set of frames for an object or scene, we first cluster their camera extrinsics (position and orientation) using $k$-means. Subsequently, we select one frame from each resulting cluster, prioritizing the frame that exhibits the fewest masked pixels (i.e., maximal object visibility within the frame). Any objects for which no valid test images remain after this selection process (e.g., due to insufficient visibility in all remaining frames not reserved for testing) are excluded from the evaluation set to maintain fairness.

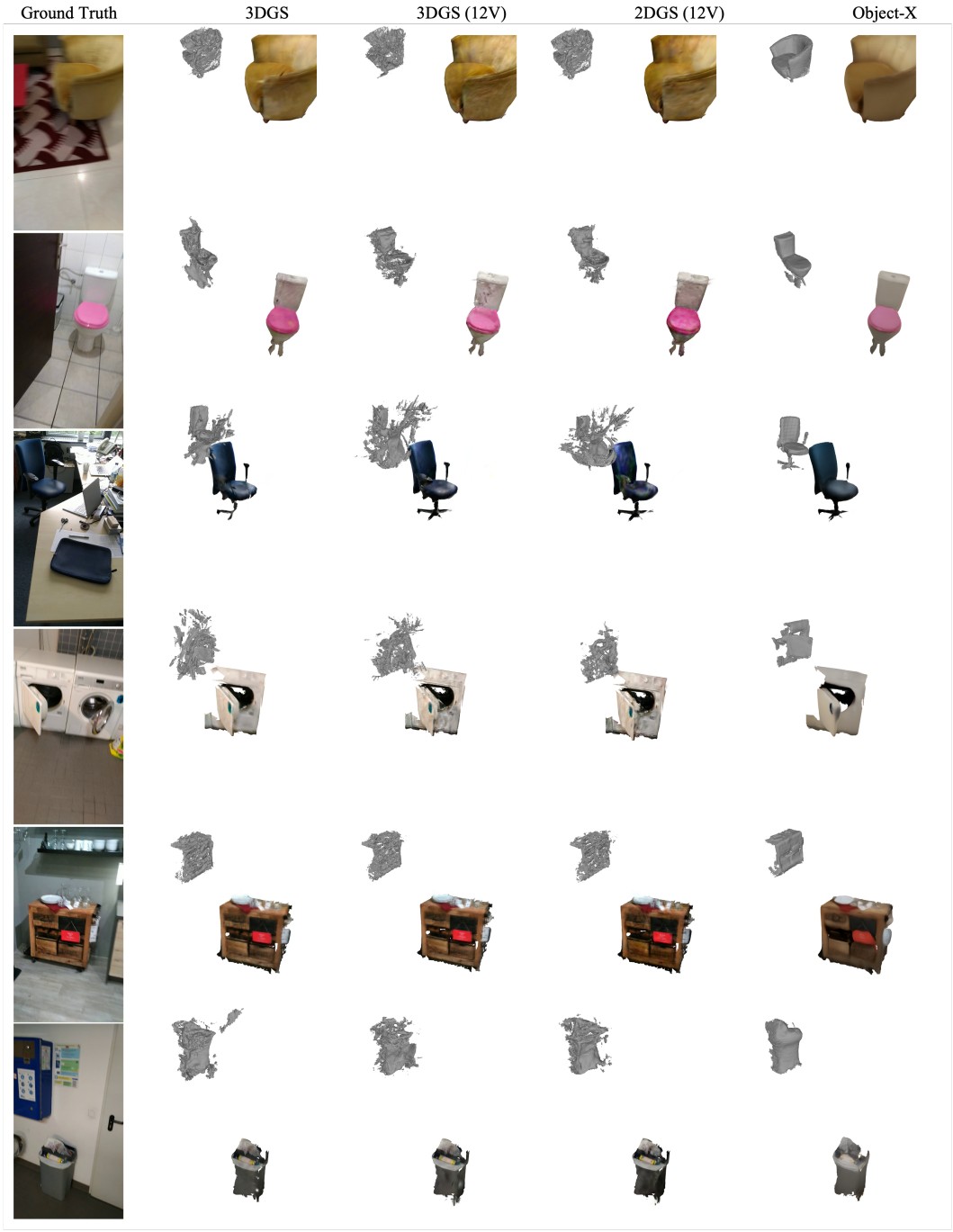

| Ground Truth | 3DGS | 3DGS (12V) | 2DGS (12V) | Object-X |

Figure 6: **Object reconstructions.** Each row shows an input object (left) and its reconstruction obtained by, from left to right: (i) 3DGS (28) optimized on all images, (ii) 3DGS or (iii) 2DGS (41) using only 12 multi-view images, and (iv) *Object-X*. For each method, we present a rendered image from the reconstructed 3D Gaussians and the corresponding mesh.

## A.2    Scene-Level Reconstruction

For full-scene evaluation, we adapt 3DGS and 2DGS to operate jointly across all objects within a scene. This is achieved by utilizing the union of the same $k$ views per object as in the object-level experiments (specifically, 12 views for 3RScan and 4 for ScanNet), but here the RGB-D

inputs are *unmasked*. Our method, *Object-X*, reconstructs the scene by independently decoding the learned U-3DGS embedding for each constituent object and then rendering their collective splats. This compositional approach requires no additional scene-level optimization. We also evaluate an augmented version, denoted as **Object-X + Opt**. This variant leverages the compositional scene from *Object-X* as an initialization for a subsequent refinement stage. Specifically, it undergoes an additional 4,000 iterations of 3DGS optimization. To ensure stability during this fine-tuning process, all learning rates are reduced by a factor of $10\times$ compared to the standard 3DGS settings.

Visualizations of scene-level reconstructions are presented in Figure 8. While *Object-X* generally produces significantly smoother results than the baseline methods, its performance can be affected by objects missing from the input segmentations (e.g., a poster on a wall, as shown in the first row of the figure, or the objects on the desk, as shown in the second row). Additionally, fine-grained details might sometimes be diminished. However, applying 3DGS optimization as a post-processing step (**Object-X + Opt**) yields substantial improvements in accuracy, effectively recovering such lost details.

### A.3  Single-Image Reconstruction

We extend our evaluation to a single-view reconstruction setting for all methods. In this scenario, 3DGS is optimized from scratch using a single masked RGB-D image (and its corresponding RGB image) for 3,000 iterations. To ensure a fair comparison, *Object-X* utilizes the same reference image. This image is selected based on criteria that maximize unmasked object coverage while minimizing cropping along the image borders. We also test our method with only RGB input with depth predicted by Metric3D (6) to generate the initial point cloud. Similarly to the scene reconstruction case, we use *Object-X* to provide an initial reconstruction which we further refine by applying an additional 1,000 iterations of 3DGS optimization, using learning rates reduced by a factor of $10\times$ (consistent with the scene-level refinement). Visual results for this setting are presented in Figure 7.

Notably, despite *Object-X* not being explicitly trained for single-image reconstruction tasks, it frequently produces visually cleaner reconstructions than 3DGS when both methods are constrained to the same single input view and 3DGS is optimized from scratch under these conditions. The reconstructed meshes are also substantially more accurate than the ones from 3DGS.

### A.4  Evaluation Summary

Across all experimental settings, methods are evaluated on a fixed set of test views, distinct from training/optimization views, on a per-object or per-scene basis as appropriate. We report standard quantitative metrics, Peak Signal-to-Noise Ratio (PSNR), and the perceptual metric LPIPS. Qualitative comparisons are provided in the relevant figures accompanying each experimental section ( e.g., Figure 6 for object-level, Figure 8 for scene-level, and Figure 7 for single-image results). Across all evaluated levels – single-view, multi-view object reconstruction, and full-scene composition – *Object-X* demonstrates strong performance, simultaneously offering significant advantages in terms of computational efficiency and flexibility in initialization.

## B  Training Details

The training procedure encompasses three primary phases: sparse representation learning, compression model training, and adaptation for downstream tasks. Each phase employs distinct optimization settings to ensure both stability and efficiency. For the sparse transformer-based encoder and decoder, we apply gradient clipping at a threshold of $0.01$. This is crucial for stabilizing the training process and preventing excessively large updates within the structured latent space. Optimization is conducted using the AdamW optimizer with a learning rate of $1 \times 10^{-4}$. This learning rate is selected to strike an effective balance between training stability and convergence speed. AdamW is chosen for its decoupled weight decay mechanism, which aids in regularizing the model without adversely affecting the gradient-based optimization updates. During the compression phase, a 3D U-Net architecture is trained to map the structured latent representation to a more compact form suitable for efficient storage or transmission. A higher learning rate of $1 \times 10^{-3}$ is utilized in this phase. This facilitates accelerated convergence while preserving reconstruction quality. Explicit gradient clipping is not

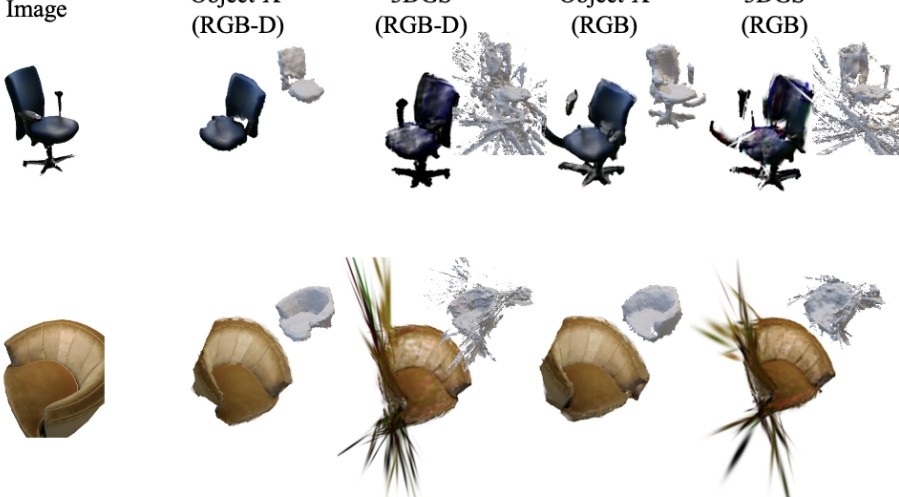

| Image | Object-X (RGB-D) | 3DGS (RGB-D) | Object-X (RGB) | 3DGS (RGB) |

Figure 7: **Qualitative comparison for image to 3D**. We compare the proposed *Object-X* to standard 3DGS (28) on RGB and RGB-D inputs. For each method, we present the image from which the object (left column) is reconstructed and the rendered novel view together with the mesh reconstructed from the 3D Gaussians by: (*2nd column*) proposed *Object-X* with RGB-D input; (*3rd*) 3DGS with RGB-D; (*4th*) proposed *Object-X* with RGB input; (*5th*) 3DGS with RGB. The proposed method leads to significantly cleaner novel views and meshes than 3DGS applied to a single image.

deemed necessary for the U-Net, as its inherent hierarchical structure and typical training dynamics provide sufficient stabilization.

For adaptation to downstream tasks, such as object localization or instance retrieval, training is performed using the AdamW optimizer with a learning rate of $1 \times 10^{-3}$ when the voxel-based latent representation is kept frozen. However, if the voxel representation is fine-tuned concurrently with the task-specific modules, a lower learning rate of $1 \times 10^{-4}$ is adopted. This approach helps to mitigate the risk of catastrophic forgetting of the learned representations. Key regularization techniques employed include the aforementioned gradient clipping and structured weight decay (e.g., as provided by the AdamW optimizer).

## C   Supplementary: Coarse Visual Localization

We provide additional details for the visual localization experiment on the 3RScan dataset. This experiment is designed to evaluate the downstream utility of the U-3DGS embeddings when augmented with auxiliary modalities.

**Training.** The model utilized for localization is trained following the auxiliary learning setup described in the main paper. We freeze the pre-trained U-3DGS encoder and decoder. Auxiliary encoders are then trained on object-level inputs derived from 3D scene graphs, specifically object relationships, attributes, and structural context. Each RGB query image is processed through a DINOv2 backbone followed by a patch-level encoder to generate patch-wise descriptors. A contrastive loss function aligns these image patches with the corresponding object embeddings within the scene. Concurrently, a compression loss ensures that the original U-3DGS component of the joint embedding remains accurately decodable. After this initial stage, all modules, including the U-3DGS components and auxiliary encoders, are jointly fine-tuned to enhance task-specific performance while preserving reconstruction fidelity.

**Setup.** Following the evaluation protocol established by SceneGraphLoc (15), we sample 123 distinct scenes from 30 rooms within the 3RScan test split. For each query image, the objective is to identify the correct scene from a candidate pool of 10 scenes, which includes the ground-truth scene. This experimental setup results in a total of *30,462 query evaluations*.

| Ground Truth | 3DGS | 3DGS (12V) | 2DGS (12V) | Object-X | Object-X + Optim |
|---|---|---|---|---|---|

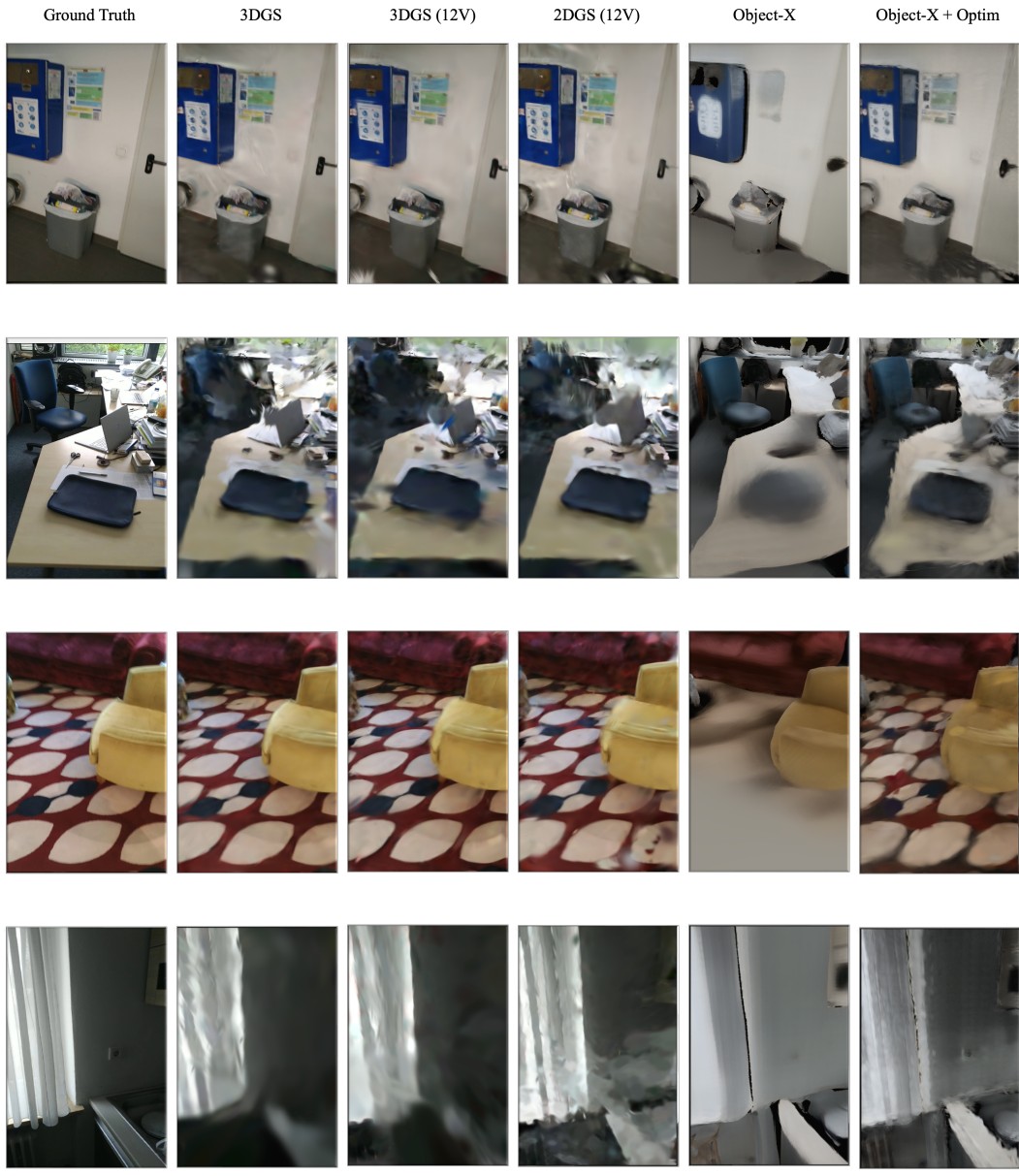

Figure 8: **Qualitative comparison for full-scene composition**. We compare the proposed *Object-X* to standard 3DGS (28) optimized on all unmasked scene images, and two 12-view baselines: 3DGS (12V) and 2DGS (12V), which optimize scenes using a subset of training images constructed by taking the union of the 12 best views selected per object.

**Evaluation.** At test time, each posed RGB image is encoded into a set of patch-level embeddings. For every candidate scene in the pool, these image patch embeddings are compared against all available object embeddings from that scene using cosine similarity. The final prediction for the scene is determined via a robust voting mechanism that aggregates all patch-object similarity scores. We report **Recall@K** for $K \in \{1, 3, 5\}$, which measures the frequency with which the correct scene appears among the top-K predicted scenes. This metric directly reflects the model's capability to localize images effectively using the learned, object-centric multimodal representation.

**Results.** We compare our results with SceneGraphLoc (15) and the recent CrossOver method (24). Our approach demonstrates competitive localization accuracy while crucially maintaining compatibility with 3D reconstruction and other downstream applications, a benefit stemming from our

jointly trained, modular representation. Detailed results are presented in Table 5. In this table, we indicate whether a given method utilizes point clouds ($\mathcal{P}$), images ($\mathcal{I}$), other modalities such as object attributes and relationships ($\mathcal{O}$), or the proposed U-3DGS embedding. Our method, leveraging U-3DGS embeddings in conjunction with other modalities, achieves the highest Recall@1 and Recall@3 scores. This outcome suggests that the proposed U-3DGS embeddings furnish information comparable to, or even richer than, that provided by raw point clouds or images for the task of visual localization.

Furthermore, we present an ablation study for our method (indicated with an asterisk * in Table 5) using only the U-3DGS embeddings, without any specific fine-tuning of our main encoder for this localization task. As anticipated, the auxiliary modalities (attributes, relationships, etc.) offer valuable complementary information, contributing significantly to the superior performance of the full model. Interestingly, even in this constrained setting (U-3DGS embeddings alone, without targeted training), our method performs comparably to the recent CrossOver approach (24). This highlights the inherent richness and suitability of our learned U-3DGS embeddings for visual localization tasks, even without explicit optimization for this specific application.

| Method | Modalities | | | | 10 scenes | | |
| --- | --- | --- | --- | --- | --- | --- | --- |
| | $\mathcal{P}$ | $\mathcal{I}$ | O | 3DGS | Recall@1 | Recall@3 | Recall@5 |
| SGLoc (15) | ✓ | ✗ | ✓ | ✗ | 53.6 | 81.9 | **92.8** |
| CrossOver (24) | ✗ | ✓ | ✗ | ✗ | 46.0 | 77.9 | 90.5 |
| *Object-X* | ✗ | ✗ | ✓ | ✓ | **56.6** | **82.2** | 91.8 |
| *Object-X** | ✗ | ✗ | ✗ | ✓ | 28.7 | 58.5 | 76.5 |
| *Object-X** | ✗ | ✗ | ✓ | ✓ | 44.8 | 72.6 | 85.7 |

Table 5: **Coarse visual localization** on the 3RScan dataset (9) using the proposed U-3DGS embedding, compared to SceneGraphLoc (15) and CrossOver (24). We report retrieval recall at 1, 3, and 5 when selecting the correct scene from 10 candidates. Evaluations are conducted using different map modalities: point cloud ($\mathcal{P}$), image ($\mathcal{I}$), other modalities ($\mathcal{O}$), and 3DGS. In the lower section (*), we also present results where the U-3DGS embedding is used without task-specific training.

# D   Supplementary: Scene Alignment

We provide additional details on the setup and evaluation procedure for the 3D Scene Alignment task on the 3RScan dataset (9), following the protocol established by SGAligner (23).

**Setup.** To construct the evaluation data, we generate sub-scenes by selecting fixed-length sequences of consecutive RGB-D frames from the 3RScan validation set. Each such sequence is then fused into a partial 3D reconstruction using volumetric integration. This process results in a total of *848 sub-scenes*, each representing a distinct viewpoint or region within an original, larger scene. From these sub-scenes, we create *1,906 pairs* by selecting pairs that originate from the same ground-truth scene. These pairs are deliberately constructed to span a wide range of spatial overlap percentages (from 10% to 90%) thereby ensuring coverage of both straightforward and challenging alignment scenarios.

**Evaluation.** We extract object embeddings independently from each sub-scene. These embeddings are produced by the same network architecture and weights trained for the scene localization task, as detailed in Section C. During the evaluation phase, we compute the *cosine similarity* between every object embedding in one sub-scene and all object embeddings in its paired sub-scene. For each object in the first sub-scene, candidate objects from the second sub-scene are ranked based on this similarity score. We evaluate the quality of these rankings using standard retrieval metrics: **Mean Reciprocal Rank (MRR)** and **Hits@K**, where $K \in \{1, 2, \ldots, 5\}$. The Hits@K metric measures the proportion of queries for which a correct match appears within the top-K ranked results, while MRR quantifies the average inverse rank of the first correct match.

# E   Supplementary: Single-Image Reconstruction

We provide qualitative results for the comparison of MIDI and Object X. The scenes on which the experiment is tested on are made available in the code.

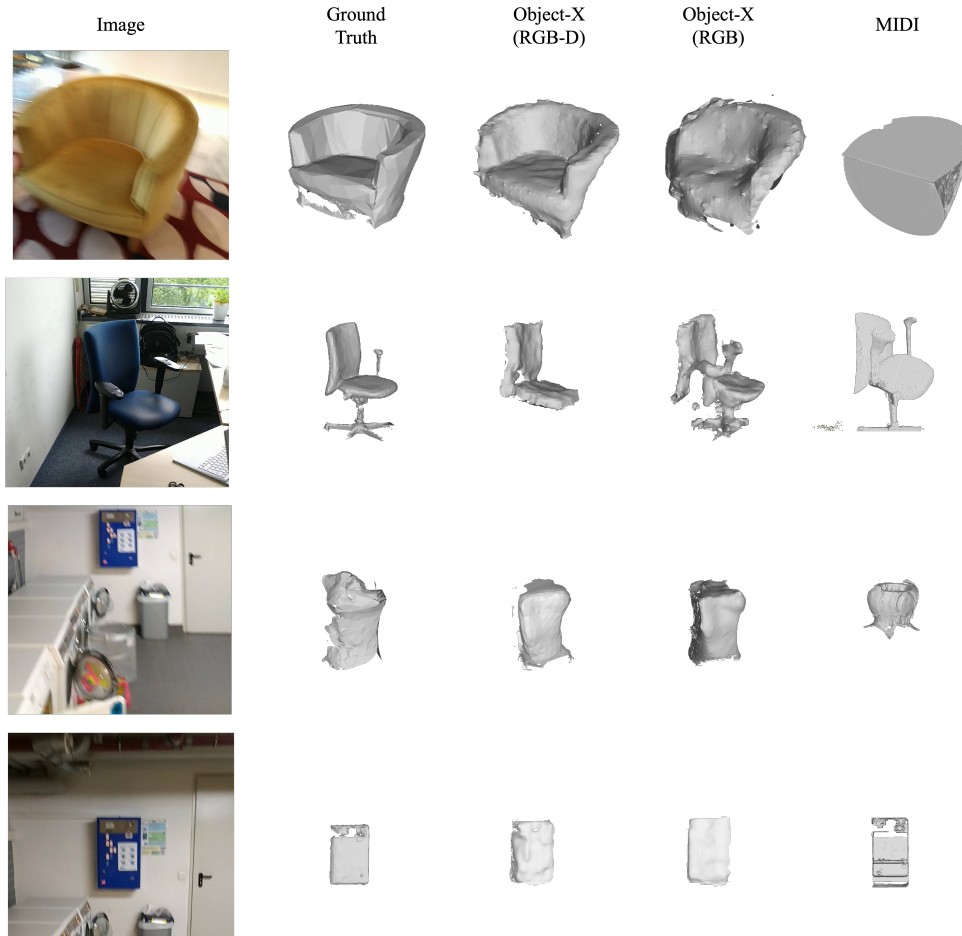

Figure 9: Qualitative comparison between MIDI (8) and Object-X on 3RScan single-image inputs. MIDI often produces realistic but misaligned shapes, while Object-X yields accurate and structurally coherent reconstructions that preserve scale and geometry.

# F   Ablation Studies

This section analyzes the impact of key components and design choices in our proposed method. Table 6 presents results as a function of the compression rate, which is defined by the resolution of the underlying voxel grid, where each voxel stores eight parameters. As a reference, we also report results for standard 3D Gaussian Splatting (3DGS). In addition to our proposed 3D U-Net architecture for compression, we evaluate a naive downsampling approach that applies max pooling followed by interpolation.

The results corresponding to a $64^3$ voxel grid resolution effectively represent our Structured Latent (SLat) representation without any subsequent compression, as this directly matches the original voxel resolution described in the main paper. The ablation results demonstrate that employing naive downsampling leads to a significant degradation in accuracy as the resolution decreases. In contrast, our proposed 3D U-Net maintains high fidelity with only a marginal loss in accuracy, while substantially reducing the number of parameters required per object from $64^3 \times 8 = 2\,097\,152$ to

a mere $8^3 \times 8 = 4\,096$. Based on this analysis, we adopt a resolution of $16^3$ for the compressed representation in all our main experiments.

Table 7 evaluates the robustness of our method to varying degrees of occlusion by systematically removing parts of an object before it is encoded. Occlusion is simulated by selecting a random point on the object's surface and removing all geometry within a sphere of diameter $d$. The diameter $d$ is defined as a fraction of the object's characteristic size; for example, $d = 0.4$ corresponds to approximately 40% of the object's volume being removed. The results indicate that even under severe occlusion, our proposed method maintains high reconstruction accuracy, thereby demonstrating its resilience to incomplete or missing input data.

| Resolution | Method | LPIPS (Mean $\pm\ \sigma$) $\downarrow$ | Median $\downarrow$ | PSNR (Mean $\pm\ \sigma$) $\uparrow$ | Median $\uparrow$ |
|---|---|---|---|---|---|
| 3DGS | - | $0.086 \pm 0.082$ | 0.060 | $30.15 \pm 5.06$ | 30.14 |
| $64^3$ (SLat) | - | $0.094 \pm 0.101$ | 0.059 | $27.30 \pm 6.13$ | 27.06 |
| $32^3$ | Naive | $0.124 \pm 0.126$ | 0.076 | $25.28 \pm 5.51$ | 25.80 |
|  | 3D Unet | $\mathbf{0.099 \pm 0.108}$ | $\mathbf{0.060}$ | $\mathbf{27.06 \pm 6.18}$ | $\mathbf{26.84}$ |
| $16^3$ | Naive | $0.189 \pm 0.137$ | 0.137 | $21.32 \pm 5.53$ | 21.41 |
|  | 3D Unet | $\mathbf{0.103 \pm 0.113}$ | $\mathbf{0.062}$ | $\mathbf{27.01 \pm 6.29}$ | $\mathbf{26.71}$ |
| $8^3$ | Naive | $0.257 \pm 0.187$ | 0.211 | $17.46 \pm 5.26$ | 16.86 |
|  | 3D Unet | $\mathbf{0.110 \pm 0.119}$ | $\mathbf{0.065}$ | $\mathbf{26.74 \pm 6.35}$ | $\mathbf{26.50}$ |

Table 6: **Ablation study on latent dimensions.** Mean and median LPIPS and PSNR on a subset of scans from the test set. We compare the standard 3DGS (as a reference), the SLat embedding without dimensionality reduction, and U-3DGS with compressed representations at $32^3$, $16^3$, and $8^3$. Also, we evaluate naive downscaling approaches using max pooling and interpolation alongside the proposed 3D U-Net. The $16^3$ resolution is selected for all other experiments as it significantly reduces storage while maintaining near-optimal reconstruction accuracy.

| $d$ | LPIPS (Mean $\pm\ \sigma$) $\downarrow$ | Median $\downarrow$ | PSNR (Mean $\pm\ \sigma$) $\uparrow$ | Median $\uparrow$ |
|---|---|---|---|---|
| 0.0 | $0.104 \pm 0.114$ | 0.062 | $26.85 \pm 6.25$ | 26.66 |
| 0.1 | $0.104 \pm 0.113$ | 0.063 | $26.96 \pm 6.32$ | 26.69 |
| 0.2 | $0.106 \pm 0.114$ | 0.064 | $26.70 \pm 6.44$ | 26.50 |
| 0.4 | $0.113 \pm 0.119$ | 0.068 | $26.07 \pm 6.70$ | 25.93 |

Table 7: **Ablation study on occlusion.** Before encoding an object, we randomly select a point on its surface and remove all parts within a spherical region of diameter $d$. For example, $d = 0.4$ corresponds to a removal region spanning 40% of the object's size. We report LPIPS and PSNR scores for different values of $d$ to assess the impact of occlusion on reconstruction quality.

