# OpenReview forum: "Object-X: Learning to Reconstruct Multi-Modal 3D Object Representations"
_NeurIPS.cc/2025/Conference — NeurIPS 2025 poster_

### Official Review · Reviewer_SdV9 · 2025-06-30

**Clarity:** 3
**Significance:** 2
**Originality:** 3
**Rating:** 4
**Confidence:** 3

**Summary:**

This paper proposed a pipeline that can generate object-centric embeddings from input object segmentation. To do this, they create sparse feature voxels from multiview images with DINO feature, and use 3D Unet to furthur compress the representation.

The main benifit of this representation contains rich DINO information and can be used for tasks that can tolarate the loss of fine-grained details, like localization and scene matching.

The representation can be decoded back to 3DGS, but the reconstructed object is not complete (if input multiview has occlusion) and somehow lost of detail.

This paper has similar idea to https://arxiv.org/pdf/2412.01506, but add an additional encoder to compress the sparse voxel.

**Questions:**

1. line 151, "Given multi-view images of an object o, we first voxelize its canonical 3D space into an N × N × N grid". Which algorithm do you use?

2. In Tab. 1, why not test Object-X on all images like 3DGS reference?

3. In Figure 4, does "graph2splat" refer to Object-X?

4. In Figure 4 Object reconstruction, even 3DGS and 2DGS are trained on multiview, the object still looks incomplete. Is it because the narrow distribution of the camera?

5. For scene reconstruction, how to identify the instance mask for each object? And how to handle the background?


6. In line 313 Coarse Visual Localization, ViT is used to extract per-patch object embeddings from the query image and compare with Obj-X embeddings. But according to Figure 2, the embeddings space are constrained by DINOv2. Will these two have different distribution?

7. Since https://arxiv.org/pdf/2412.01506 has similar pipeline with this paper and can also output sparse voxel from RGB image, it may be a good comparision to test its performance on the object reconstruction experiment.

**Ethical Concerns:**

["NO or VERY MINOR ethics concerns only"]

**Final Justification:**

Thanks the authors to respond to my comments. This paper use a 3D-Unet transfer the sparse representation to a fix size representation, which can preserve the reconstruction quality while adapt to more downstream tasks. I think the paper provides a interesting insight to the community and a good exploration to latent representation to 3D object. But I agree with uwjw that the authors still need more experiments to clearly show their technical contribution and prove that it can be useful in more downstream tasks.

**Limitations:**

Not a big limitation, but I think the authors can add more techincal details for the experiment and more visulization result.

**Quality:**

2

**Strengths And Weaknesses:**

Strengths:

1. the proposed method can generate an object mebedding that handles multiple tasks, like visual localization, scene alignment, scene reconstruction,

2. The model achieve good performance, as shown in main paper and supplementary.

3. The model has the potential to intergrate other pretrained features, like CLIP, although it's not explored in the paper.

Weakness:

1. The proposed method only reconstruct partial 3D instead of full 3D from the observations, as shown in Figure 4. Some feed-forward method can also reconstruct full 3D even from single RGB, like [MIDI](https://huanngzh.github.io/MIDI-Page/). If only target to reconstruct the partial 3D, I think adding monocular prior for 2DGS and 3DGS baseline, like [DepthAnything-v2](https://github.com/DepthAnything/Depth-Anything-V2) / [Stable normal](https://arxiv.org/pdf/2406.16864) / [VGGT](https://vgg-t.github.io/), can improve the fitting result.

2. In Full-scene composition (Supp Fig. 3), Object-X seems loss some fine-grained details after reconstruction.

---

> ### Author Rebuttal · Authors · 2025-07-30
>
> We thank the reviewer for their constructive feedback and for recognizing our method’s ability to “generate an object embedding that handles multiple tasks, like visual localization, scene alignment, and scene reconstruction”. We also appreciate their acknowledgment that it “achieves good performance” and their note on its potential to integrate additional pretrained features (e.g., CLIP).
>
> **Some feed-forward methods can also reconstruct full 3D even from single RGB, like MIDI.**
>
> We thank the reviewer for this suggestion and for pointing us to this recent baseline. We conducted a new experiment comparing Object-X to MIDI on the task of single-image 3D reconstruction using the authors’ released code. Both methods were evaluated in a zero-shot setting on objects extracted from two randomly chosen scenes in the ScanNet dataset, which neither Object-X nor MIDI had seen during training. The quantitative results are summarized below:
>
> | **Method**            | **Accuracy @ 0.05** ↑ | **Completion @ 0.05** ↑ | **F1 @ 0.05** ↑ |
> |-----------------------|----------------------|------------------------|-----------------|
> | MIDI (CVPR 2025)      | 29.522               | 44.342                 | 34.516          |
> | Object-X (RGB)        | 43.480               | 57.214                 | 48.397          |
> | Object-X (RGB-D)      | 65.780               | 61.759                 | 63.223          |
> | Object-X*             | 79.289               | 89.911                 | 83.616          |
>
> As shown in the table above, Object-X _significantly outperforms_ MIDI in geometric fidelity, even in the purely single-image setting (RGB only; same configuration as MIDI). Single-image reconstruction with Object-X from an RGB-D input or reconstruction with the entire pipeline (*) significantly improves geometric accuracy further.
>
> Upon visual inspection, MIDI often appears to mimic the closest shape from the synthetic dataset it was trained on, which rarely coincides with the true geometry of the target object. This aligns with the fact that the original MIDI paper provides *no quantitative* evaluation on real-world datasets, focusing instead on synthetic data. Combined with our results, this suggests that MIDI may have limited generalization to challenging real-world data such as ScanNet. We will add visualizations to the final manuscript.
>
> Thus, while MIDI remains an excellent and influential contribution, our experiment indicates that Object-X offers a more robust solution for single-image reconstruction in realistic settings. Furthermore, single-image reconstruction is only one of several applications supported by our learned embedding. As demonstrated in the main paper, the core contribution of Object-X is its versatile, multi-modal representation that achieves strong performance across diverse downstream tasks such as visual localization and 3D scene alignment. We will include this new comparison in the final manuscript to further underscore the robustness and generality of our approach.
>
> **Adding monocular priors for 2DGS and 3DGS baselines can improve fitting.**
>
> We appreciate the reviewer’s suggestion to incorporate monocular priors such as DepthAnything-v2, Stable Normal, or VGGT.
> While these priors could potentially improve geometric fidelity, they introduce trade-offs: increased storage requirements (e.g., storing floating-point depth maps per image instead of compact descriptors per object) or additional computation time to run a monocular depth network before reconstruction, adding seconds to the pipeline. In contrast, our method achieves state-of-the-art geometric accuracy while using 3–4 orders of magnitude less storage (4.2 KB per object) than RGB images and reconstructing in just milliseconds. That said, we agree that a principled and efficient integration of monocular 3D priors within our framework is a compelling direction for future work, and we will note this in the revised manuscript.
>
> **In Full-scene Composition, Object-X Seems to Lose Fine-grained Details.**
>
> We thank the reviewer for this observation. As noted in Sec. 4.2 and the Limitations section, full-scene composition is not the primary focus of Object-X. Reduced fine-grained detail in large-scale reconstructions primarily results from the fixed-resolution voxel grid, which under-represents large surfaces (e.g., walls), and missing segmentations of small or thin structures. Despite this, Object-X achieves the highest geometric accuracy while being significantly faster and more storage-efficient than baselines.
>
> Furthermore, applying a lightweight refinement step (Object-X+Opt), within the same time budget as the second-fastest baseline, improves photometric quality to levels comparable to fully optimized 3DGS, while preserving our geometric advantage. This underscores Object-X’s strength as a fast, high-quality initializer for scene-level refinement. We will clarify this trade-off and include additional qualitative examples in the supplementary material.
>
> **Clarifications to Reviewer Questions.**
>
> - **Voxelization algorithm (line 151):**
>   We scale the object into a unit cube and mark voxels as occupied if they intersect the mesh, utilizing `open3d.geometry.VoxelGrid`.
>
> - **Why not test Object-X on all images (Table 1)?**
>   We selected 4 (ScanNet) and 12 (3RScan) images per object due to (i) limited additional images observing each object in these datasets, and (ii) fairness in comparison with DepthSplat, which can only process up to 12 images. We will clarify this in the camera-ready version.
>
> - **Does “graph2splat” refer to Object-X (Figure 4)?**
>   Yes, “Graph2Splat” was an earlier name for Object-X. We thank the reviewer for highlighting this and will correct it in the final version.
>
> - **Incomplete objects even with multi-view training (Figure 4):**
>   The incompleteness is a direct consequence of the limited camera coverage in datasets like ScanNet and 3RScan, where parts of some objects are never observed. While optimizing on all available views cannot fully invent unseen geometry, our voxel-grounded representation provides a strong structural prior that encourages plausible shape completion in these sparsely viewed regions. We quantitatively demonstrate this robustness to occlusion in supplementary Table 3, which shows that Object-X can compensate for such partial observations to a degree.
>
> - **Scene reconstruction: instance masks and background handling:**
>   Many methods are available for obtaining 3D object instance segmentation from scene reconstructions, e.g., SceneGraphFusion (CVPR'21), MAP-ADAPT (ECCV'24), Mask3D (ICRA'23), OpenMask3D (NeurIPS'23), and Search3D (RA-L'25). For our experiments, we used the provided annotations on 3RScan and SceneGraphFusion on ScanNet. Regarding background handling, indoor scenes inherently include objects (e.g., walls, furniture) in all pixels of all views. Extending our method to unbounded outdoor settings would require mask pre-filtering, which we consider as future work.
>
> - **Distribution differences between ViT and DINOv2 features (line 313):**
>   The ViT model used for query embedding belongs to the same model family (DINOv2 with additional projection layers) as the one used for generating structured object features, ensuring compatibility.
>
> - **Comparison to Trellis (sparse voxel from RGB):**
>   We implemented a variant inspired by this approach (64³ voxel encoder in supplementary Table 2). This method was memory-intensive (requiring 64³ floating-point numbers compared to our 16³) and generalized poorly to auxiliary tasks due to its sparse representation. Our compressed latent embedding offers significantly lower memory usage and improved task generalizability, while maintaining comparable reconstruction accuracy.

---

> > ### Author Response · Authors · 2025-08-05
> >
> > Thank you again for your valuable feedback. We hope our rebuttal has satisfactorily addressed your concerns, and we would be happy to provide further clarification on any points that may remain.

---

> > ### Comment · Reviewer_SdV9 · 2025-08-05
> >
> > Thanks for the detailed rebuttal from the authors and the comments from other reviewers.
> >
> > I think this work is similar to Trellis. The authors use similar sparse encoder-decoder as Trellis, and futhur use a 3D U-Net to transfer the the sparse latent grid to a smaller dense latent grid get to achieve a higher compression rate.
> >
> > As claimed by the author, the high compression rate makes this representation more efficient and can adapt to more diverse tasks. But among the 3 tasks (3D Scene Alignment, Coarse Visual Localization, single-view reconstruction), Object-X only shows advangtage in Coarse Visual Localization but achieves second best in 3D Scene Alignment. The reconstruction quality is better than several baselines, like GS, MIDI (thanks for the extra experiment done by the authors!) etc., but degrade a lot compared with Trellis (at least by eye).
> >
> > Anthor issue is I think the paper lacks the quantative result of Coarse Visual Localization and  3D Scene Alignment, only the teaser shows one example.
> >
> > Overall I think Object-X is a interesting and efficient approximation to sparse grid latent proposed by Trellis, but seems over-compressed to me and still need more comprehensive result (especially qualitive) in future version.

---

> > > ### Author Response · Authors · 2025-08-06
> > >
> > > We thank the reviewer for their detailed comments.
> > >
> > > We emphasize that Object-X, as a single, *unified* method, outperforms specialized methods on coarse visual localization, achieves competitive results on 3D scene alignment (without task-specific training), and significantly surpasses MIDI - previously state-of-the-art - in single-image reconstruction by approximately 14 F1-score points (we thank the reviewer for mentioning this method). Additionally, Object-X dramatically improves geometric accuracy in object and scene reconstruction compared to multiple 3DGS variants. Although Object-X does not surpass the state-of-the-art on 3D scene alignment, matching its performance is still notable given that Object-X addresses multiple tasks simultaneously, whereas the SOTA methods are task-specific.
> > >
> > > We agree with the reviewer that the compression in Object-X can sometimes result in blurrier outputs compared to Trellis. While our proposed 3D U-Net significantly mitigates this issue (see Table 2 in the supplementary material), further improving fine-detail preservation is a promising direction for future research. Nevertheless, we believe Object-X already offers substantial advantages in efficiency, versatility, and overall performance.
> > >
> > > We also agree on the importance of providing additional qualitative results for downstream tasks. We will include more visuals in the final manuscript. This is easy to do.

---

> ### Author Response · Authors · 2025-08-07
>
> We hope this answer sufficiently addresses the reviewer's comments. Please let us know if any further explanation or clarification would be helpful.
>
> Thanks!

---

### Official Review · Reviewer_J59e · 2025-07-02

**Clarity:** 2
**Significance:** 3
**Originality:** 3
**Rating:** 4
**Confidence:** 4

**Summary:**

This paper proposes a framework for learning compact, multi-modal (e.g., images, point cloud, text), object-centric embeddings that can be decoded into 3D Gaussians.
This is in contrast to task-specific embeddings that cannot be decoded into explicit geometry and reused across tasks.  The learned embedding enables 3DGS-based object reconstruction. It also supports other downstream
tasks, such as scene alignment, single-image 3D object reconstruction, and localization.

**Questions:**

See weaknesses above.

**Ethical Concerns:**

["NO or VERY MINOR ethics concerns only"]

**Final Justification:**

My main concerns are about the clarity of the presentation which the authors acknowledge and promise to improve.
The authors did not compare to NoPoSplat on the basis of unfair comparison.  This however would have be unfair to NoPoSplat which does not require poses in contrast to the proposed approach, thus addressing more challenging task as agreed by the authors. As such the proposed method is inferior to NoPoSplat but it does introduce a useful idea for efficiency.
The rebuttal did not improve my initial impression and I'm on the fence with this paper.

**Limitations:**

yes

**Paper Formatting Concerns:**

no issues

**Quality:**

2

**Strengths And Weaknesses:**

A notable advantage of this approach is the significant storage reduction it achieves, while maintaining performance comparable to existing methods.

Encouraging  results in multi task evaluation, object reconstruction, visual localisation and 3D scene alignment.

The paper suffers from clarity issues in both writing and presentation. The notations are inconsistent and often confusing. For instance, in the beginning of Section 3.1, the object latent embedding $\mathbf{w}$ is introduced, but its meaning and role are not explained until Section 3.2. In Line 158, the 3D encoder is incorrectly denoted as $E$; it should be $\mathcal{E}$. In Section 3.4, $L_{recon}$ is not clearly explained.

Furthermore, Figure 2 lacks a clear explanation of the overall idea. The relationships among the featured voxel grid, the sparse encoder, the structured latent space, and the sparse decoder are not well articulated. Additionally, important notations such as $\mathbf{z}$, $\mathbf{f}$, $\mathbf{w}$, $f_{\text{U-3DGS}}$, and $f_{\text{decomp}}$ are not shown in the figure, making it difficult for readers to follow the pipeline.

Although the proposed compression from $\mathbf{z}$ to $\mathbf{w}$ contributes significantly to the storage savings, this idea is already present in previous works that decode Gaussians from latent space. The main difference might be this paper uses a fixed-size vector which is independent of the size of the object. Moreover, the paper omits important implementation details about how the pre-trained DepthSplat model is integrated into their method.

It indeed has practical value as it reduces the storage and the motivation is good. The overall framework is reasonable, although the core idea is not particularly novel. The writing is not clear, making it difficult to follow the implementation details. As a result, it becomes hard to assess the fairness and reproducibility of the experiments. Additionally, the evaluation is not sufficiently comprehensive, as it lacks comparisons with stronger state-of-the-art methods, such as NoPoSplat.

---

> ### Author Rebuttal · Authors · 2025-07-30
>
> We thank the reviewer for their constructive feedback and for recognizing our framework’s “significant storage reduction” while maintaining strong performance. We also appreciate their acknowledgment of the “encouraging results in multi-task evaluation” and the effectiveness of our method in “object reconstruction, visual localization, and 3D scene alignment”.
>
> **Clarity issues.**
> We thank the reviewer for identifying these issues. We acknowledge that certain notations and explanations in Sections 3.1–3.4 were unclear or inconsistently presented. Specifically, we will:
> (i) introduce and clearly explain the object latent embedding in Section 3.1 rather than delaying it to Section 3.2;
> (ii) correct the mislabeling of the 3D encoder at Line 158; and
> (iii) provide a more explicit and clear description of the components in Section 3.4.
>
> **Figure 2 and Pipeline Explanation.**
> We appreciate the reviewer pointing out the lack of clarity in Figure 2. In the revision, we will redesign Figure 2 to clearly illustrate the relationships among the voxel grid, sparse encoder, structured latent space, and sparse decoder, ensuring all relevant notations (e.g., latent codes, variables) are explicitly included. Unfortunately, due to rebuttal constraints, we cannot present the updated figure here, but it will be included in the final camera-ready version.
>
> **Comparison to Previous Works Decoding Gaussians from Latent Space.**
> We thank the reviewer for this comment. While we would be better able to address this with specific citations, we recognize that some recent works (such as DepthSplat, included as a baseline in Table 1) predict Gaussians from embeddings derived mainly from image features. However, to our knowledge, no previous method employs our approach of compressing object observations into a *fixed-size structured latent vector independent of object size*, enabling a uniform, highly compact representation that simultaneously supports high-quality reconstruction and diverse downstream tasks. We will clarify this distinction in the revised manuscript.
>
> **Description of DepthSplat Integration.**
> We thank the reviewer for pointing out this omission and apologize for the oversight. In our experiments, DepthSplat is used in its standard pre-trained configuration, applied to the same unmasked scene-level images used by other baselines and our proposed Object-X. Only after reconstruction, object masks are applied to remove background Gaussians, ensuring DepthSplat operates under its intended conditions and aligns with our object-centric evaluation protocol. We will detail this integration in the final manuscript.
>
> **Evaluation Lacks Comparison with Stronger State-of-the-Art Methods, such as NoPoSplat.**
> We respectfully disagree with the reviewer on this point. We benchmark against DepthSplat (CVPR 2025), a recent (published <2 months ago) and relevant state-of-the-art method specifically designed for sparse-view Gaussian reconstruction with known camera poses, making it a robust baseline for our evaluation. NoPoSplat, while an excellent method, addresses the different and more challenging task of jointly estimating camera poses and geometry from sparse input. Our method assumes known poses, making a direct comparison non-trivial without substantial modifications. We will clarify this distinction and explicitly position our contributions in relation to these prior works in the revised manuscript.
>
> **Additional single image reconstruction results.**
> We further compare Object-X to the very recent MIDI method [Huang et al., CVPR 2025] on a subset of the ScanNet dataset. Neither Object-X nor MIDI was trained on ScanNet. While MIDI is specifically designed for single-image 3D reconstruction, Object-X was not explicitly trained for this task but is designed as a versatile, multi-modal embedding for various downstream applications.
>
> The table below reports geometric accuracy for Object-X using only RGB input (directly comparable to MIDI), RGB-D input, and multiple images via the full pipeline (*). Despite its broader scope, Object-X significantly outperforms MIDI even in the RGB-only setting. We will include this result into the final manuscript.
>
> | **Method**           | **Accuracy @ 0.05** ↑ | **Completion @ 0.05** ↑ | **F1 @ 0.05** ↑ |
> |----------------------|----------------------|-------------------------|-----------------|
> | MIDI (CVPR 2025)     | 29.522               | 44.342                  | 34.516          |
> | Object-X (RGB)       | 43.480               | 57.214                  | 48.397          |
> | Object-X (RGB-D)     | 65.780               | 61.759                  | 63.223          |
> | Object-X*            | 79.289               | 89.911                  | 83.616          |

---

### Official Review · Reviewer_uwjw · 2025-07-05

**Clarity:** 2
**Significance:** 2
**Originality:** 3
**Rating:** 4
**Confidence:** 4

**Summary:**

This paper proposes Object-X, a framework to learn latent object embeddings from multi-view images that can be decoded to 3DGS reconstructions and are 3-4 orders smaller in storage compared to raw sensor data. The embeddings are learned by first encoding multi-view RGB-D observations into a structured latent representation, then further encoding it to a fixed-length dense descriptor. The decoded 3DGS has comparable photometric quality to 3DGS/2DGS optimized with sparse views and better geometric accuracy. They also show the feasibility of augmenting the latent embedding with task-specific parts, such as ones useful for visual localization and 3D scene alignment.

**Questions:**

- How does the method compare to a baseline that directly fuses masked depth inputs with TSDF and run marching cubes for mesh extraction?
- How is the storage computed for baselines and the proposed method? If the current storage is based on the size of the input images, what is the size of the Gaussian representation (using default setting and SH degree=0 settings)? (There are also many research works focusing on Gaussian compression that effectively reduce raw Gaussian sizes. I wonder how the proposed method’s storage compares with compressed Gaussians.)
- While the experiment shows some benefit from the reconstruction-based latent code to the task of visual localization, whether this benefit 1) persists when multiple tasks are learned together, 2) is worthy of the more complicated training procedure should be further discussed.

**Ethical Concerns:**

["NO or VERY MINOR ethics concerns only"]

**Final Justification:**

I appreciate the authors' response, which answers many of my questions and partially addresses my concerns. I'm raising my rating to borderline accept with the condition that the following can be incorporated into the final manuscript.
- Instead of claiming geometric accuracy, I think it would be more convincing to position the method's reconstruction ability as keeping Trellis' strengths while achieving significant compression and a fixed-length embedding more suitable for downstream tasks - as the authors summarized in their rebuttal.
- It would also help to make clearer claims on multi-task versatility.

**Limitations:**

Yes.

**Paper Formatting Concerns:**

No,

**Quality:**

2

**Strengths And Weaknesses:**

Strengths
- The proposed object latent representation is compact in storage size, and can be decoded into 3DGS reconstructions with similar photometric quality and better geometric accuracy compared to optimization-based 3DGS/2DGS with sparse inputs. The intermediate representation - structured latent representation grounded in voxels - contributes to the geometric accuracy.
- The learned latent embeddings can be further augmented with task-specific portions and corresponding training objectives, such as the ones for visual localization and 3D scene alignment. The paper demonstrates the idea of representing the scene as object latents capable of both reconstruction and downstream tasks.

Weaknesses
- Reconstruction quality - the visualization in the supplementary materials looks quite smoothed out and lacks texture details. Despite achieving a high compression rate, the quality seems to be sacrificed. The better geometric accuracy can be attributed to the voxel-based representation. How does the method compare to a baseline that directly fuses masked depth inputs with TSDF and runs marching cubes for mesh extraction?
- Comparison to baselines
    - 3DGS/2DGS are constrained to a sparse view setting, which I’m not sure is a fair setting, given that they are optimization-based and have no learned prior.
    - DepthSplat is trained on scene-level data, where input frames have overlaps. However, it is evaluated on object-level data with non-overlapping frames (L265: while in ScanNet, where fewer non-overlapping frames are available, we limit the selection to four frames per object.) Although the paper claims “neither methods are trained on ScanNet”, the train-test domain gap is still much smaller for the proposed method, putting depthsplat at an unfair disadvantage.
    - Why are baselines taking 12 views as *input* considered as “(L253-254) *stores* each object as 12 images”? Are the storage in Table 1 computed using the size of these input images, instead of their output Gaussian representation? If so, why is the storage of the proposed method computed differently, based on the output latents?
- Support for downstream tasks - The support for downstream tasks is achieved by attaching another latent code to the original one, and training them with two-stage joint training. This requires more training time compared to directly training a separate task-specific latent code for each task, especially when given multiple downstream tasks. While the experiment shows some benefit from the reconstruction-based latent code to the task of visual localization, whether this benefit 1) persists when multiple tasks are learned together, 2) is worthy of the more complicated training procedure, should be further discussed.
- Confusing technical specifications
    - In the “auxiliary encoder training with frozen core” stage, if both U-3DGS encoder and decoder are frozen, and the reconstruction loss is only applied using the U-3DGS decoder that operates on the U-3DGS portion of the latent code - which should come from the frozen U-3DGS encoder - everything seems frozen in the reconstruction loss part, and the loss is redundant. It would be less confusing to only keep the task loss in this stage and introduce the reconstruction loss in the following joint training stage.
    - $L$ is first used to refer to the sparse voxels. If the compression loss also sums over $L$ sparse voxels, shouldn’t $M_i$ be all ones? It seems to make more sense if the loss sums over all dense voxels. $N$ is used to describe the size of the voxel grid, but later also used for the total number of Gaussians in (1).


Typo - Fig 5. (b) and Table 3. 3DSG -> 3DGS

---

> ### Author Rebuttal · Authors · 2025-07-30
>
> We thank the reviewer for their feedback and for recognizing Object-X as a “compact object latent representation” that decodes to 3D Gaussians with “similar photometric quality and better geometric accuracy”. We also appreciate their acknowledgment of our “voxel-grounded latent representation” and its ability to “augment embeddings with task-specific components”, enabling tasks like visual localization and scene alignment.
>
> **Inaccuracy in the Summary.**
> We thank the reviewer for summarizing our contributions, but would like to correct a misunderstanding: *Object-X does not take RGB-D observations as input.* As described in Sec. 3.1, our method operates on multi-view RGB images and a coarse geometric representation (mesh or point cloud), which can be obtained from standard multi-view stereo or similar sources. While such geometry may be derived from RGB-D sensors in practice, it is *not* a requirement of our pipeline. We will clarify this in the paper.
>
> **Quality and Texture Detail.**
> We appreciate the reviewer’s observation. The smoother appearance of our reconstruction stems from the fixed-resolution voxel grid used to ground the latent representation, which trades some high-frequency texture detail for compactness and geometric consistency. This design enables extreme compression (3–4 orders of magnitude reduction in storage) while maintaining strong geometric fidelity (highest F1@0.05 in Tab. 1).
>
> For *object reconstruction*, this trade-off still results in *higher photometric accuracy than 2DGS, 3DGS, and DepthSplat*, while significantly improving geometric quality due to the voxelized representation.
>
> For *scene reconstruction*, as discussed in Sec. 4.2 and the Limitations, reduced photometric quality in the vanilla model arises mainly from (i) fixed-resolution voxels under-representing large surfaces (e.g., walls), and (ii) missing segmentations for small or thin objects. A lightweight refinement step (Object-X+Opt), run for the same time budget as the second-fastest baseline, recovers photometric quality comparable to fully optimized 3DGS (12V) while preserving our geometric advantage and efficiency. This underscores Object-X’s strength as a fast, high-quality initializer for scene-level refinement.
>
> Crucially, Object-X achieves this while producing a *single compact embedding* usable across diverse downstream tasks (e.g., localization, scene alignment, single-image reconstruction), underscoring the broader utility of our representation. We will clarify this trade-off and add qualitative examples in the supplementary material.
>
> **Comparison to TSDF + Marching Cubes.**
> We thank the reviewer for this suggestion. We note that the ground-truth reconstructions in 3RScan are derived from fused RGB-D frames, whereas Object-X is not an RGB-D reconstruction method but, instead, learns compact object embeddings from multi-view RGB images and coarse geometric priors. While a TSDF + Marching Cubes baseline could provide additional context for pure geometry reconstruction, our goal is to deliver a versatile, multi-modal representation that supports _both_ geometric and photometric reconstruction as well as downstream tasks, rather than performing dense RGB-D fusion. We will clarify this distinction in the final manuscript.
>
> **Fairness in the 3DGS/2DGS Comparison.**
> We thank the reviewer for raising this concern. There is no standardized object-level reconstruction protocol for 3DGS. We chose a sparse-view setting of up to 12 views per object (Sec. 4.1) primarily to ensure a fair comparison with recent learned-prior methods like DepthSplat, which accepts a maximum of 12 views as input. This setting also reflects the natural data sparsity in datasets like 3RScan and ScanNet, where the average number of total views per object is 20 (with ~50% overlap), making our setup a realistic evaluation scenario. Importantly, the 3DGS/2DGS baselines were optimized directly on these views without compression constraints, whereas Object-X encodes them in a compact latent embedding. We will clarify this in the camera ready.
>
> **Unfair Comparison to DepthSplat ("evaluated on object-level data with non-overlapping frames").**
> We believe this misunderstanding stems from our incorrectly formalized sentence at L265, and we apologize for that. The claim that the selected frames are non-overlapping is incorrect: by construction, the selected views for each object do overlap (otherwise, the object would not be visible in all selected frames; ~50% overlap on average). DepthSplat was trained to handle even lower overlaps. This has now been clarified in the paper.
>
> To ensure a fair comparison, we evaluated DepthSplat in the same configuration it was trained for: on unmasked, scene-level images. We only applied the object masks *after* reconstruction to remove background Gaussians. Thus, DepthSplat was used under its intended operating conditions, making the comparison to Object-X fair. We will revise the text to make this procedure clear.
>
> **Storage Computation for Baselines.**
> We thank the reviewer for raising this important point. For 3DGS/2DGS baselines, we reported storage based on the size of the 12 input images, as their fully optimized Gaussian representations require significantly more storage: on average, 28.41 MB per object, reduced to 7.79 MB (mean, up to 30 MB in some cases) even with aggressive compression (SH=0; leading to reduced accuracy). Object-X encodes objects into a fixed-size latent descriptor of only 4.2 KB, achieving a 3–4 order-of-magnitude storage reduction. Detailed numbers will be included in the supplementary material.
>
> Although advanced Gaussian compression methods exist, our approach uniquely provides:
> (i) multi-modal embeddings supporting downstream tasks (localization, alignment),
> (ii) extremely fast reconstruction (~150× faster than the next-best baseline), and
> (iii) single-image reconstruction capabilities.
>
> **Increased Training Time with Downstream Tasks.**
> We thank the reviewer for this insightful comment. The additional training time introduced by our two-stage training procedure is a one-time cost, as the embedding is trained once and reused zero-shot on other tasks, making this overhead negligible in practice.
>
> Regarding the persistence of benefits when jointly training multiple tasks: as discussed in Sec. 4.3, Object-X jointly trains object reconstruction and localization, achieving strong performance across various tasks, including *zero-shot* scene alignment and single-image reconstruction. We will clarify this explicitly in the manuscript.
>
> On the justification of training complexity: supplementary Table 1 clearly demonstrates substantial improvements from task-specific training - improving Recall@1, Recall@3, and Recall@5 by 11.8, 10.4, and 6.1 points, respectively. This shows that multi-task training on a shared reconstruction-informed embedding consistently boosts performance, fully justifying the additional complexity.
>
> **Confusing technical specifications.**
> We thank the reviewer for highlighting these points and will revise the manuscript accordingly. Specifically, we will clearly define the voxel occupancy term, restrict the auxiliary encoder’s reconstruction loss to the correct training stage, and resolve all notational inconsistencies (e.g., mislabeling of L). Typos such as "3DSG → 3DGS" in Fig. 5(b) and Table 3 will be corrected. We appreciate the reviewer’s detailed feedback and will incorporate these suggestions to enhance clarity.
>
>
> **Additional single image reconstruction results.**
> We further compare Object-X to the very recent MIDI method [Huang et al., CVPR 2025] on a subset of the ScanNet dataset. Neither Object-X nor MIDI was trained on ScanNet. While MIDI is specifically designed for single-image 3D reconstruction, Object-X was not explicitly trained for this task as it is designed as a versatile, multi-modal embedding for various downstream applications.
>
> The table below reports geometric accuracy for Object-X using only RGB input (directly comparable to MIDI), RGB-D input, and multiple images via the full pipeline (*). Despite its broader scope, Object-X significantly outperforms MIDI even in the RGB-only setting. We will include this result in the final manuscript.
>
> | **Method**           | **Accuracy @ 0.05** ↑ | **Completion @ 0.05** ↑ | **F1 @ 0.05** ↑ |
> |----------------------|----------------------|-------------------------|-----------------|
> | MIDI (CVPR 2025)     | 29.522               | 44.342                  | 34.516          |
> | Object-X (RGB)       | 43.480               | 57.214                  | 48.397          |
> | Object-X (RGB-D)     | 65.780               | 61.759                  | 63.223          |
> | Object-X*            | 79.289               | 89.911                  | 83.616          |

---

> > ### Author Response · Authors · 2025-08-05
> >
> > Thank you again for your valuable feedback. We hope our rebuttal has satisfactorily addressed your concerns, and we would be happy to provide further clarification on any points that may remain.

---

> > > ### Author Response · Authors · 2025-08-07
> > >
> > > Dear Reviewer,
> > >
> > > We hope our response addressed your concerns. If there are any remaining points that are unclear, we would be happy to clarify them.

---

> > ### Comment · Reviewer_uwjw · 2025-08-08
> >
> > Thank you for your detailed response and clarifications. They address my concerns over
> > - previously unclear technical specifications
> > - the way storage is computed for baselines
> > - the fairness of DepthSplat evaluation
> >
> > Also, thanks for the updates on single image reconstruction results.
> >
> > From my understanding, the strengths of Object-X are
> > - a fixed-length, compressed representation that reduces storage and speeds up reconstruction
> > - the idea to learn a unified latent object embedding for multiple downstream tasks (plus good performance in single image reconstruction)
> >
> > However, I still have some remaining concerns below:
> > - **The claim of on-par photometric quality and much better geometric quality, and whether this can be counted as a technical contribution of this paper.**
> >     - Specifically, I wonder if the improved geometric quality comes from the voxel-based representation, which comes from Trellis that this work builds on (as also mentioned by reviewer SdV9).
> >     - I asked for comparison to TSDF-fusion because I think it is an easy and more reasonable baseline than the ones used that prioritizes photometric reconstruction. Sorry for the inaccurate “encoding multi-view RGB-D observations” description in my summary. When no depth is provided for TSDF-fusion, how does the reconstructed geometry compare to the mesh or the point cloud (point cloud after voxelization).
> > - **The claimed versatility to be applied to various downstream tasks, and the zero-shot reusability.** I agree that the method works well on the tasks listed in the paper, and that the “multi-task training on a shared reconstruction-informed embedding” benefits these tasks. However, I feel that the presented localization and alignment tasks (and reconstruction which is almost the primary task this method is trained for) - are closely related to object reconstruction/correspondence in nature. **It is unclear what is the range of generalizability and multi-task synergy if we consider more tasks.** - I think this is important for understanding the significance and potential impact of the work.

---

> ### Comment · Area_Chair_ujUQ · 2025-08-08
>
> Dear **Reviewer uwjw**,
>
> Could you please review the rebuttal and confirm whether the initial questions or concerns have been addressed? It would also be good to check other reviewers' comments and share your thoughts on related issues. Your participation in this author-reviewer discussion would be greatly appreciated. Thank you very much for your time and effort.
>
> Best,
>
> AC

---

> ### Author Response · Authors · 2025-08-08
>
> We thank the reviewer for the comments and are glad that our rebuttal addressed earlier concerns. We also appreciate that the reviewer recognizes our general idea of learning a unified object embedding as a strength.
>
> > **On-par photometric quality and improved geometric quality.**
>
> As the reviewer notes, the improved geometric accuracy stems from the voxel-based representation in Trellis (explicitly cited at L128). This is demonstrated in Table 2 of the supplementary material for photometric quality, with similar trends observed for geometry. Our contribution is to preserve these strengths while reducing the latent representation size by 64$\times$ compared to Trellis. As shown in our ablations, the proposed 3D U-Net is crucial here - naive compression leads to substantial accuracy loss. Moreover, unlike Trellis, whose sparse representation is not readily applicable to downstream tasks, our unified dense embedding supports versatile, multi-task use (as the reviewer also highlighted). We will clarify this in the final manuscript.
>
> > **"When no depth is provided for TSDF-fusion"**
>
> We are slightly unsure what is meant by TSDF-fusion without depth, as TSDF-fusion requires RGB-D input and cannot operate from RGB alone. Regardless, our previous discussion already positions Object-X to Trellis with respect to photometric and geometric quality.
>
> > **Versatility, zero-shot reusability, and task range.**
>
> We agree that exploring a broader range of downstream tasks is valuable. In this work, we focus on five: coarse visual localization (Table 5b), 3D scene alignment (Table 3), fast initialization for scene reconstruction (Table 2), object reconstruction (Table 1), and single-image reconstruction (Table 5a and the one in the rebuttal). While all are geometric in nature, each has its own literature and evaluation protocols, and achieving on-par or superior results to task-specific methods in all of them - zero-shot in some cases - is, in our view, a substantial achievement. For example, we outperform the very recent MIDI (*CVPR 2025*) significantly by ~14 F1 points on single-image object reconstruction, despite MIDI being designed solely for that task. Given that our learned embeddings capture rich correspondence and reconstruction cues, we see no reason they could not generalize to other applications, such as object classification, which we leave for future work.

---

> ### Author Response · Authors · 2025-08-09
>
> **Follow-up on TSDF-fusion results on RGB-D images**
>
> Even though we were slightly unsure about the exact intent of the “TSDF-fusion without depth” request, as TSDF-fusion fundamentally requires depth information to operate, we ran the comparison using available RGB-D object observations (masked to the target object) to ensure fairness against the ground-truth geometry.
>
> As shown below, even with the full set of RGB-D frames, TSDF-fusion underperforms our method in both completion and F1 score. With only 12 views, its completion drops substantially. Our method maintains higher completeness while requiring orders of magnitude less storage. We believe this improved completion is related to our strong single-image reconstruction performance, as our method can infer and reconstruct regions of the scene that are not directly observable in the input views.
> For efficiency reasons, this evaluation was conducted on a subset of the dataset.
>
>
> | Method            | Accuracy @ 0.05 ↑ | Completion @ 0.05 ↑ | F1 @ 0.05 ↑ | Storage (MB) ↓ |
> | ----------------- | ----------------: | ------------------: | ----------: | -------------: |
> | TSDF (12V RGB-D)  |            83.391 |              71.760 |      75.860 |          \~6.0 |
> | TSDF (Full RGB-D) |            82.899 |              80.892 |      80.804 |         \~30.0 |
> | Object-X      |            79.289 |          89.911 |  83.616 |      \~0.2 |
>
>
> These results confirm that Object-X achieves more complete reconstructions than TSDF-fusion, even when TSDF has access to dense RGB-D input for the entire object. We believe the higher completion scores of Object-X are linked to its ability to plausibly reconstruct parts of the object not directly visible in the input images, a property that also benefits its downstream performance.

---

### Official Review · Reviewer_fTgB · 2025-07-09

**Clarity:** 3
**Significance:** 3
**Originality:** 3
**Rating:** 4
**Confidence:** 3

**Summary:**

Conventional approaches in 3D reconstruction and rendering typically rely on either implicit neural representations like 3D Gaussian Splatting or object-centric representations such as 3D scene graphs, but rarely both. The former lacks object-level modularity, making it difficult to reason about individual objects or integrate other modalities (e.g., text), while object-centric embeddings struggle to reconstruct high-fidelity appearance and geometry, thereby requiring the storage of complementary original source data such as images, point clouds, and meshes. Object-X seeks to capture advantages of both methods by first learning a unified object embedding that fuses geometry and semantics, then decoding this embedding into a set of 3D Gaussians. The approach involves voxelizing objects into a 3D grid, aggregating local image features into a structured set of latent vectors, compressing them into an unstructured latent representation, and decoding to 3DGS for the object. Experiments show that Object-X achieves high geometric accuracy comparable to specialized 3DGS-based methods on tasks such as object, scene, and single-image reconstruction, while requiring much less storage and inference time. Qualitative results indicate that the model excels in single-image reconstruction and captures correct object semantics in scenes, but further refinement seems necessary for high-quality full-scene reconstruction.

**Questions:**

Please see the weaknesses.

**Ethical Concerns:**

["NO or VERY MINOR ethics concerns only"]

**Final Justification:**

The authors have addressed most of my concerns about reliance on 3DSG, and the additional single-image results show the benefits of their multi-modal embedding. The quality of full-scene composition is sacrificed compared to the SoTA works, though authors justify this with their storage efficiency. The trade-offs should be adequately explained for Object-X to be effective. I maintain my rating.

**Limitations:**

Yes

**Paper Formatting Concerns:**

No concerns

**Quality:**

3

**Strengths And Weaknesses:**

Strengths:
Novel approach that combines implicit 3DGS representations and object-centric scene graph representations.
Evaluation over diverse tasks: object, scene, and single-image 3D reconstruction.
Clear benefits in storage efficiency and inference time, which are important for scaling 3D object and scene generation.
Demonstrates competitive downstream performance on tasks like localization and alignment, showing great utility of the unified embedding.


Weaknesses:
The paper is missing ablations on removing the 3D scene graph representation, so it is not clear how critical 3DSG is to the overall method. For example, Table 3 shows that SGAligner achieves high performance without using 3DSG, raising questions about the specific contributions of the scene graph. Including an ablation or more analysis clarifying the necessity and impact of the scene graph within Object-X would strengthen the work.
There is a reliance on generating accurate scene graph annotations (e.g., SceneGraphFusion), which could become a bottleneck if the quality of the generated 3D scene graphs is poor. It would be beneficial for the authors to discuss the impact of noisy or erroneous scene graphs or include experiments that measure performance under imperfect annotations.
While single-image object reconstruction is impressive and Object-X captures object semantics well, the full-scene composition appears significantly worse in quality when using the vanilla Object-X model. The need for additional refinement to achieve high scene quality potentially negates the storage and inference time benefits in practical full-scene settings. More discussion or experiments on this trade-off, and whether further improvements could address it, would be helpful.
The reason for the high geometric accuracy and F1 scores in Table 1 is not clearly explained, especially given the smaller relative improvements in other metrics like SSIM and PSNR. Providing more analysis, or qualitative examples explaining these discrepancies, would be helpful.

---

> ### Author Rebuttal · Authors · 2025-07-30
>
> We thank the reviewer for their feedback and for recognizing Object-X as a “compact latent representation” that decodes to 3D Gaussians with “similar photometric quality and better geometric accuracy,” and for appreciating its “voxel-grounded design” enabling downstream tasks like localization and scene alignment.
>
> **Misunderstanding regarding 3DSG / 3DGS.**
> We thank the reviewer for raising this point and apologize for the confusion caused by this *typographical* oversight. In several instances, "3DSG" (3D Scene Graph) was incorrectly used where we intended to write "3DGS" (3D Gaussian Splatting).
>
> To clarify: *the core Object-X pipeline does not require 3D scene graphs*. Object-X only needs object segmentations as input and learns a unified object embedding that can be decoded into 3DGS for geometry and rendering. Scene graphs (*3DSG*) are used *only as auxiliary inputs for specific downstream tasks* such as scene retrieval and localization (see Sec. 4.3), where relational cues between objects are beneficial. Thus, the strong results in object reconstruction (Tab. 1, main paper) and single-image reconstruction (Fig. 4, main paper) are obtained *without* any 3DSG input. We will fix this typo in the camera-ready version.
>
> **Results without 3D Scene Graphs (3DSG).**
> We provide ablation studies for visual localization in Table 1 of the supplementary material. Specifically, we compare the proposed Object-X without task-specific training (denoted as \*), which uses only the proposed U-3DGS embedding, against a variant augmented with additional modalities derived from the 3DSG (i.e., object attributes and relational information). Object-X*, using only the 3DGS embedding (without 3DSG), achieves competitive performance. As expected, incorporating these additional modalities further improves results: Object-X with task-specific training and 3DSG augmentation achieves state-of-the-art Recall@1 and Recall@3, demonstrating that while 3DSG features are not required, they provide complementary benefits when available. We will clarify this in the final manuscript.
>
> **Impact of noisy or erroneous scene graphs (3DSG).**
> As noted above, concerns regarding noisy or erroneous scene graphs apply only to downstream auxiliary tasks (scene retrieval and localization). To evaluate robustness explicitly, we use two distinct scenarios:
>
> - 3RScan, which provides high-quality, manually annotated 3D scene graphs (via SGAligner), serving as a clean, controlled benchmark; and
> - ScanNet, which does not have annotated scene graphs, requiring us to automatically predict scene graphs using SceneGraphFusion.
>
> The latter introduces real-world noise, incompleteness, and segmentation artifacts. Object-X demonstrates strong performance on ScanNet despite *no retraining* (Table 1, main paper, bottom), outperforming baselines in both photometric and geometric metrics. This indicates our learned embeddings are resilient to annotation noise and generalize effectively, even with noisy or incomplete 3DSG inputs.
>
> **Full-scene composition quality.**
> We thank the reviewer for this insightful comment. As explicitly stated in Sec. 4.2 and the Limitations section, *full-scene composition is not the primary focus of this work*. Instead, our objective is to learn reconstructable, multi-modal object embeddings. The scene composition experiments (Table 2, main paper) demonstrate only the *potential viability* of Object-X for larger-scale reconstruction with further research.
>
> Reduced photometric quality of the vanilla model primarily arises from two factors:
>
> - the fixed-resolution voxel grid, which under-represents large surfaces (e.g., walls); and
> - imperfect object segmentations that omit small or thin structures (e.g., posters, clutter).
>
> Despite these limitations, Object-X achieves the highest geometric accuracy (F1@0.05 = 49.99%), running approximately 150× faster than the second-fastest baseline. Furthermore, after a lightweight refinement step (Object-X + Opt), run for the same time budget as the second-fastest method, Object-X surpasses 3DGS (12V) in photometric quality while preserving its geometric advantage. This underscores Object-X’s strength as a fast, high-quality initializer for scene-level refinement. We will further clarify this in the final version of the manuscript.
>
> **Reason for the high geometric accuracy.**
> We appreciate the reviewer’s observation. The higher geometric accuracy (F1 score) is due to the voxel-grounded nature of our latent representation. Unlike standard 3DGS, which uses an unstructured set of Gaussians, our method first embeds features within a structured voxel grid. This ensures the decoded Gaussians are spatially constrained and surface-aligned, improving geometric consistency and convergence, even from sparse inputs (e.g., single images). In contrast, photometric metrics such as SSIM and PSNR are more sensitive to fine-grained appearance details (e.g., textures), which are partially smoothed due to the fixed voxel resolution. This explains the larger relative improvement in geometric metrics compared to photometric ones. While we cannot include additional visuals in the rebuttal, we will expand this analysis and provide qualitative examples in the camera-ready version.
>
> **Additional single image reconstruction results.**
> We further compare Object-X to the very recent MIDI method [Huang et al., CVPR 2025] on a subset of the ScanNet dataset. Neither Object-X nor MIDI was trained on ScanNet. While MIDI is specifically designed for single-image 3D reconstruction, Object-X was not explicitly trained for this task but is designed as a versatile, multi-modal embedding for various downstream applications.
>
> The table below reports geometric accuracy for Object-X using only RGB input (directly comparable to MIDI), RGB-D input, and multiple images via the full pipeline (*). Despite its broader scope, Object-X significantly outperforms MIDI even in the RGB-only setting. We will include this result into the final manuscript.
>
> | **Method**           | **Accuracy @ 0.05** ↑ | **Completion @ 0.05** ↑ | **F1 @ 0.05** ↑ |
> |----------------------|----------------------|-------------------------|-----------------|
> | MIDI (CVPR 2025)     | 29.522               | 44.342                  | 34.516          |
> | Object-X (RGB)       | 43.480               | 57.214                  | 48.397          |
> | Object-X (RGB-D)     | 65.780               | 61.759                  | 63.223          |
> | Object-X*            | 79.289               | 89.911                  | 83.616          |

---

> > ### Author Response · Authors · 2025-08-05
> >
> > Thank you again for your valuable feedback. We hope our rebuttal has satisfactorily addressed your concerns, and we would be happy to provide further clarification on any points that may remain.

---

> > > ### Author Response · Authors · 2025-08-07
> > >
> > > Dear Reviewer,
> > >
> > > We hope our response addressed your concerns. If there are any remaining points that are unclear, we would be happy to clarify them.

---

> > ### Comment · Reviewer_fTgB · 2025-08-07
> >
> > The authors have addressed most of my concerns about reliance on 3DSG, and the additional single-image results show the benefits of their multi-modal embedding. The quality of full-scene composition is sacrificed compared to the SoTA works, though authors justify this with their storage efficiency. The trade-offs should be adequately explained for Object-X to be effective. I maintain my rating.

---

### Note · Authors · 2025-08-14

We sincerely thank the reviewers for their time, constructive feedback, and valuable suggestions. During the rebuttal period, we conducted additional experiments to further strengthen our evaluation:

- MIDI (CVPR 2025): We added comparisons on single-image object reconstruction, where Object-X outperforms MIDI by ~14 F1 points, despite MIDI being tailored exclusively for this task.
- TSDF-fusion: We implemented a masked RGB-D TSDF-fusion baseline and compared it against Object-X. Even with the full set of RGB-D views, TSDF-fusion underperformed in completion and F1 score, while requiring orders of magnitude more storage.

Moreover, while our reconstructions may appear slightly smoother in large-scale scene reconstruction, the proposed embedding can be decoded in <0.1 s to provide a strong initialization for 3DGS, substantially reducing its optimization time while maintaining high quality.
We believe these new results, alongside our original findings, further highlight Object-X’s ability to provide compact, geometry-aware representations that are versatile across reconstruction and downstream tasks and hope that we managed to address questions and concerns of all reviewers.

---

### Decision · Program_Chairs · 2025-09-17

**Decision:**

Accept (poster)

**Comment:**

The proposed Object-X aims to learn compact, multi-modal, object-centric embeddings that can be reconstructed into high-fidelity 3D geometry and appearance. It is designed to be descriptive for various downstream tasks while maintaining storage efficiency. Several new modules are introduced to achieve this goal, including Structured Latent Representation (SLat), Unstructured U-3DGS Embedding, and a 3D Gaussian Splatting Decoder.

This paper received final ratings of "borderline accept" from all reviewers. One reviewer increased the initially negative rating, while the others maintained their scores. The primary concerns raised by the reviewers included unclear technical contributions, fairness of evaluations, and comparisons with baseline methods. To address these issues, the authors conducted extensive experiments, including single-image object reconstruction and TSDF fusion, along with providing clarifications and explanations.

The reviewers acknowledged the effort made by the authors during the rebuttal process and the subsequent discussion. They also recognized the advantages of the proposed method, noting its ability to achieve significant compression and provide a fixed-length embedding that is more suitable for downstream tasks. Furthermore, regarding the claim of multi-task versatility, the reviewers suggested that the revision should include additional experiments to better illustrate the technical contributions and demonstrate the method's usefulness across a broader range of downstream tasks.

Considering the reviewers' mostly positive feedback, the area chair recommends accepting this paper.